# Improved Analysis and Rates for Variance Reduction under Without-replacement Sampling Orders

**Xinmeng Huang**[*]
University of Pennsylvania
Philadelphia, PA 19104
xinmengh@sas.upenn.edu

**Kun Yuan**[*]
DAMO Academy, Alibaba Group
Bellevue, WA 98004
kun.yuan@alibaba-inc.com

**Xianghui Mao**
Tsinghua University
Beijing, China 100084
xianghui.xh.mao@gmail.com

**Wotao Yin**
DAMO Academy, Alibaba Group
Bellevue, WA 98004
wotao.yin@alibaba-inc.com

## Abstract

When applying a stochastic algorithm, one must choose an order to draw samples. The practical choices are without-replacement sampling orders, which are empirically faster and more cache-friendly than uniform-iid-sampling but often have inferior theoretical guarantees. Without-replacement sampling is well understood only for SGD without variance reduction. In this paper, we will improve the convergence analysis and rates of variance reduction under without-replacement sampling orders for composite finite-sum minimization.

Our results are in two-folds. First, we develop a damped variant of Finito called Prox-DFinito and establish its convergence rates with random reshuffling, cyclic sampling, and shuffling-once, under both convex and strongly convex scenarios. These rates match full-batch gradient descent and are state-of-the-art compared to the existing results for without-replacement sampling with variance-reduction. Second, our analysis can gauge how the cyclic order will influence the rate of cyclic sampling and, thus, allows us to derive the optimal fixed ordering. In the highly data-heterogeneous scenario, Prox-DFinito with optimal cyclic sampling can attain a sample-size-independent convergence rate, which, to our knowledge, is the first result that can match with uniform-iid-sampling with variance reduction. We also propose a practical method to discover the optimal cyclic ordering numerically.

## 1 Introduction

We study the finite-sum composite optimization problem

$$\min_{x \in \mathbb{R}^d} F(x) + r(x) \quad \text{and} \quad F(x) = \frac{1}{n} \sum_{i=1}^{n} f_i(x). \tag{1}$$

where each $f_i(x)$ is differentiable and convex, and the regularization function $r(x)$ is convex but not necessarily differentiable. This formulation arises in many problems in machine learning [34, 39, 14], distributed optimization [20, 3, 19], and signal processing [4, 9].

The leading methods to solve (1) are first-order algorithms such as stochastic gradient descent (SGD) [28, 2] and stochastic variance-reduced methods [14, 6, 7, 17, 10, 32]. In the implementation of

---

[*]Equal Contribution. Correspondence to: Kun Yuan

35th Conference on Neural Information Processing Systems (NeurIPS 2021).

Table 1: Number of individual gradient evaluations needed by each algorithm to reach an $\epsilon$-accurate solution. Notation $\tilde{\mathcal{O}}(\cdot)$ hides logarithmic factors. Error metrics $(\mathbb{E})\|\nabla F(x)\|^2$ and $(\mathbb{E})\|x - x^\star\|^2$ are used for convex and strongly convex problems, respectively.

| Algorithm | Supp. Prox | Sampling | Asmp$^\diamond$ | Convex | Strongly Convex |
|---|---|---|---|---|---|
| Prox-GD | Yes | Full-batch | $F(x)$ | $\mathcal{O}(n\frac{L^2}{\epsilon})$ | $\mathcal{O}(n\frac{L}{\mu}\log(\frac{1}{\epsilon}))$ |
| SGD [22] | No | Cyclic | $f_i(x)$ | $\mathcal{O}(n(\frac{L}{\epsilon})^{\frac{3}{2}})$ | $\mathcal{O}(n(\frac{L}{\mu})^{\frac{3}{2}}\frac{1}{\sqrt{\epsilon}})$ |
| SGD [22] | No | RR | $f_i(x)$ | $\mathcal{O}(n^{\frac{1}{2}}(\frac{L}{\epsilon})^{\frac{3}{2}})$ | $\mathcal{O}(\sqrt{n}(\frac{L}{\mu})^{\frac{3}{2}}\frac{1}{\sqrt{\epsilon}})$ |
| PSGD [21] | Yes | RR | $f_i(x)$ | $\searrow$ | $\mathcal{O}(n(\frac{L}{\mu})^{\frac{3}{2}}\frac{1}{\sqrt{\epsilon}})$ |
| PIAG [35] | Yes | Cyclic/RR | $F(x)$ | $\searrow$ | $\mathcal{O}(n\frac{L}{\mu}\log(\frac{1}{\epsilon}))$ |
| AVRG [37] | No | RR | $f_i(x)$ | $\searrow$ | $\mathcal{O}(n(\frac{L}{\mu})^2\log(\frac{1}{\epsilon}))$ |
| SAGA [33] | Yes | Cyclic | $f_i(x)$ | $\mathcal{O}(n^3\frac{L^2}{\epsilon})$ | $\mathcal{O}(n^3(\frac{L}{\mu})^2\log(\frac{1}{\epsilon}))$ |
| SVRG [33] | Yes | Cyclic | $f_i(x)$ | $\mathcal{O}(n^3\frac{L^2}{\epsilon})$ | $\mathcal{O}(n^3(\frac{L}{\mu})^2\log(\frac{1}{\epsilon}))$ |
| DIAG [23] | No | Cyclic | $f_i(x)$ | $\searrow$ | $\mathcal{O}(n\frac{L}{\mu}\log(\frac{1}{\epsilon}))$ |
| Cyc. Cord. Upd [5] | Yes | Cyclic/RR | $f_i(x)$ | $\mathcal{O}(n2^n\frac{L^2}{\epsilon})$ | $\mathcal{O}(n(\frac{L}{\mu})^3\log(\frac{1}{\epsilon}))$ |
| SVRG [18]♣ | No | Cyclic/RR | $F(x)$ | $\mathcal{O}(n\frac{L^2}{\epsilon})$ | $\mathcal{O}(n(\frac{L}{\mu})^{\frac{3}{2}}\log(\frac{1}{\epsilon}))$ |
| SVRG [18]♣ | No | RR | $F(x)$ | $\mathcal{O}(n\frac{L^2}{\epsilon})$ | $\mathcal{O}(n\frac{L}{\mu}\log(\frac{1}{\epsilon}))^\dagger$ |
| SVRG [18]♣ | No | RR | $f_i(x)$ | $\mathcal{O}(n\frac{L^2}{\epsilon})$ | $\mathcal{O}(n^{\frac{1}{2}}(\frac{L}{\mu})^{\frac{3}{2}}\log(\frac{1}{\epsilon}))^\dagger$ |
| Prox-DFinito (Ours) | Yes | RR | $f_i(x)$ | $\mathcal{O}(n\frac{L^2}{\epsilon})$ | $\mathcal{O}(n\frac{L}{\mu}\log(\frac{1}{\epsilon}))$ |
| Prox-DFinito (Ours) | Yes | worst cyc. order | $f_i(x)$ | $\tilde{\mathcal{O}}(n\frac{L^2}{\epsilon})$ | $\tilde{\mathcal{O}}(n\frac{L}{\mu}\log(\frac{1}{\epsilon}))$ |
| Prox-DFinito (Ours) | Yes | best cyc. order | $f_i(x)$ | $\tilde{\mathcal{O}}(\frac{L^2}{\epsilon})^\ddagger$ | $\tilde{\mathcal{O}}(n\frac{L}{\mu}\log(\frac{1}{n\epsilon}))^\ddagger$ |

$^\diamond$ The "Assumption" column indicates the scope of smoothness and strong convexity, where $F(x)$ means smoothness and strong convexity in the average sense and $f_i(x)$ assumes smoothness and strong convexity for each summand function.

♣ [18] is an **independent** and **concurrent** work.

$^\dagger$ This complexity is attained under big data regime: $n > \mathcal{O}(\frac{L}{\mu})$.

$^\ddagger$ This complexity can be attained in a highly heterogeneous scenario, see details in Sec 4.2.

these methods, each $f_i(x)$ can be sampled either *with* or *without replacement*. Without-replacement sampling draws each $f_i(x)$ exactly *once* during an epoch, which is numerically faster than with-replacement sampling and more cache-friendly; see the experiments in [1, 38, 11, 7, 37, 5]. This has triggered significant interests in understanding the theory behind without-replacement sampling.

Among the most popular without-replacement approaches are cyclic sampling, random reshuffling, and shuffling-once. Cyclic sampling draws the samples in a cyclic order. Random reshuffling reorders the samples at the beginning of each sample epoch. The third approach, however, shuffles data only once before the training begins. Without-replacement sampling have been extensively studied for SGD. It was established in [1, 38, 11, 22, 24] that without-replacement sampling enables SGD with faster convergence For example, it was proved that without-replacement sampling can speed up uniform-iid-sampling SGD from $\tilde{\mathcal{O}}(1/k)$ to $\tilde{\mathcal{O}}(1/k^2)$ (where $k$ is the iteration) for strongly-convex costs in [11, 12], and $\mathcal{O}(1/k^{1/2})$ to $\mathcal{O}(1/k)$ for the convex costs in [24, 22]. [31] establishes a tight lower bound for random reshuffling SGD. Recent works [27, 22] close the gap between upper and lower bounds. Authors of [22] also analyzes without-replacement SGD with non-convex costs.

In contrast to the mature results in SGD, variance-reduction under without-replacement sampling are less understood. Variance reduction strategies construct stochastic gradient estimators with vanishing gradient variance, which allows for much larger learning rate and hence speed up training process. Variance reduction under without-replacement sampling is difficult to analyze. In the strongly convex scenario, [37, 33] provide linear convergence guarantees for SVRG/SAGA with random reshuffling, but the rates are worse than full-batch gradient descent (GD). Authors of [35, 23] improved the rate so that it can match with GD. In convex scenario, existing rates for without-replacement sampling with variance reduction, except for the rate established in an independent and concurrent work [18],

are still far worse than GD [33, 5], see Table 1. Furthermore, no existing rates for variance reduction under without-replacement sampling orders, in either convex or strongly convex scenarios, can match those under uniform-iid-sampling which are essentially sample-size independent. There is a clear gap between the known convergence rates and superior practical performance for without-replacement sampling with variance reduction.

## 1.1 Main results

This paper narrows such gap by providing convergence analysis and rates for proximal DFinito, a proximal damped variant of Finito/MISO [7, 17, 26], which is a well-known variance reduction algorithm, under without-replacement sampling orders. Our main achieved results are:

- We develop a proximal damped variant of Finito/MISO called Prox-DFinito and establish its gradient complexities with *random reshuffling, cyclic sampling, and shuffling-once*, under both convex and strongly convex scenarios. All these rates match with gradient descent, and are state-of-the-art (up to logarithm factors) compared to existing results for without-replacement sampling with variance-reduction, see Table 1.

- Our novel analysis can gauge how a cyclic order will influence the rate of Prox-DFinito with cyclic sampling. This allows us to identify the *optimal* cyclic sampling ordering. Prox-DFinito with optimal cyclic sampling, in the highly data-heterogeneous scenario, can attain a *sample-size-independent* convergence rate, which is the first result, to our knowledge, that can match with uniform-iid-sampling with variance reduction in certain scenarios. We also propose a numerical method to discover the optimal cyclic ordering cheaply.

## 1.2 Other related works

Our analysis on cyclic sampling is novel. Most existing analyses unify random reshuffling and cyclic sampling into the same framework; see the SGD analysis in [11], the variance-reduction analysis in [10, 36, 23, 37], and the coordinate-update analysis in [5]. These analyses are primarily based on the "sampled-once-per-epoch" property and do not analyze the orders within each epoch, so they do not distinguish cyclic sampling from random reshuffling in analysis. [16] finds that random reshuffling SGD is basically the average over all cyclic sampling trials. This implies cyclic sampling can outperform random reshuffling with a well-designed sampling order. However, [16] does not discuss how much better cyclic sampling can outperform random reshuffling and how to achieve such cyclic order. Different from existing literatures, our analysis introduces an order-specific norm to gauge how cyclic sampling performs with different fixed orders. With such norm, we are able to clarify the worst-case and best-case performance of variance reduction with cyclic sampling.

Simultaneously and independently, a recent work [18] also provided an improved rates for variance reduction under without-replacement sampling orders that can match with gradient descent. However, [18] does not discuss whether and when variance reduction with replacement sampling can match with uniform sampling. In addition, [18] studies SVRG while this paper studies Finito/MISO. The convergence analyses in these two works are very different. The detailed comparison between this work and [18] can be referred to Sec. 3.3.

## 1.3 Notations

Throughout the paper we let $\text{col}\{x_1, \cdots, x_n\}$ denote a column vector formed by stacking $x_1, \cdots, x_n$. We let $[n] := \{1, \cdots, n\}$ and define the proximal operator as

$$\mathbf{prox}_{\alpha r}(x) := \arg\min_{y \in \mathbb{R}^d} \{\alpha\, r(y) + \frac{1}{2}\|y - x\|^2\} \tag{2}$$

which is single-valued when $r$ is convex, closed and proper. In general, we say $\mathcal{A}$ is an operator and write $\mathcal{A} : \mathcal{X} \to \mathcal{Y}$ if $\mathcal{A}$ maps each point in space $\mathcal{X}$ to another space $\mathcal{Y}$. So $\mathcal{A}(\boldsymbol{x}) \in \mathcal{Y}$ for all $\boldsymbol{x} \in \mathcal{X}$. For simplicity, we write $\mathcal{A}\boldsymbol{x} = \mathcal{A}(\boldsymbol{x})$ and $\mathcal{A} \circ \mathcal{B}\boldsymbol{x} = \mathcal{A}(\mathcal{B}(\boldsymbol{x}))$ for operator composition.

**Cyclic sampling.** We define $\pi := (\pi(1), \pi(2), \ldots, \pi(n))$ as an arbitrary *determined* permutation of sample indexes. The order $\pi$ is *fixed* throughout the entire learning process under cyclic sampling.

**Random reshuffling.** When starting each epoch, a *random* permutation $\tau := (\tau(1), \tau(2), ..., \tau(n))$ is generated to specify the order to take samples. Let $\tau_k$ denote the permutation of the $k$-th epoch.

---

**Algorithm 1** Prox-DFinito

---

**Input:** $\bar{z}^0 = \frac{1}{n}\sum_{i=1}^{n} z_i^0$, step-size $\alpha$, and $\theta \in (0,1)$;
  **for** epoch $k = 0, 1, 2, \cdots$ **do**
    **for** iteration $t = kn+1, kn+2, \cdots, (k+1)n$ **do**
      $x^{t-1} = \mathbf{prox}_{\alpha r}(\bar{z}^{t-1})$;
      Pick $i_t$ with some rule;
      Update $z_{i_t}^t$ and $\bar{z}^t$ according to (4a) and (5);
    **end for**
    $z_i^{(k+1)n} \leftarrow (1-\theta)z_i^{kn} + \theta z_i^{(k+1)n}$ for any $i \in [n]$;    ▷ a damping step
    $\bar{z}^{(k+1)n} \leftarrow (1-\theta)\bar{z}^{kn} + \theta\bar{z}^{(k+1)n}$;           ▷ a damping step
  **end for**

---

## 2 Proximal Finito/MISO with Damping

The proximal gradient method to solve problem (1) is

$$z_i^t = x^{t-1} - \alpha\nabla f_i(x^{t-1}), \quad \forall\, i \in [n] \tag{3a}$$

$$x^t = \mathbf{prox}_{\alpha r}\Big(\frac{1}{n}\sum_{i=1}^{n} z_i^t\Big) \tag{3b}$$

To avoid the global average that passes over all samples, we propose to update one $z_i$ per iteration:

$$z_i^t = \begin{cases} x^{t-1} - \alpha\nabla f_i(x^{t-1}), & i = i_t \\ z_i^{t-1}, & i \neq i_t \end{cases} \tag{4a}$$

$$x^t = \mathbf{prox}_{\alpha r}\Big(\frac{1}{n}\sum_{i=1}^{n} z_i^t\Big). \tag{4b}$$

When $i_t$ is invoked with uniform-iid-sampling and $r(x) = 0$, algorithm (4a)–(4b) reduces to Finito/MISO [7, 17]. When $i_t$ is invoked with cyclic sampling and $r(x) = 0$, algorithm (4a)–(4b) reduces to DIAG [23] and WPG [19]. We let $\bar{z}^t := \frac{1}{n}\sum_{i=1}^{n} z_i^t$. The update (4a) yields

$$\bar{z}^t = \bar{z}^{t-1} + (z_{i_t}^t - z_{i_t}^{t-1})/n. \tag{5}$$

This update can be finished with $\mathcal{O}(d)$ operations if $\{z_i^t\}_{i=1}^{n}$ are stored with $\mathcal{O}(nd)$ memory. Furthermore, to increase robustness and simplify the convergence analysis, we impose a damping step to $z_i$ and $\bar{z}$ when each epoch finishes. The proximal damped Finito/MISO method is listed in Algorithm 1. Note that the damping step does not incur additional memory requirements. A more practical implementation of Algorithm 1 is referred to Algorithm 3 in Appendix A.

### 2.1 Fixed-point recursion reformulation

Algorithm (4a)–(4b) can be reformulated into a fixed-point recursion in $\{z_i\}_{i=1}^{n}$. Such a fixed-point recursion will be utilized throughout the paper. To proceed, we define $\boldsymbol{z} = \mathrm{col}\{z_1, \cdots, z_n\} \in \mathbb{R}^{nd}$ and introduce the average operator $\mathcal{A} : \mathbb{R}^{nd} \to \mathbb{R}^d$ as $\mathcal{A}\boldsymbol{z} = \frac{1}{n}\sum_{i=1}^{n} z_i$. We further define the $i$-th block coordinate operator $\mathcal{T}_i : \mathbb{R}^{nd} \to \mathbb{R}^{nd}$ as

$$\mathcal{T}_i\boldsymbol{z} = \mathrm{col}\{z_1, \cdots, (I - \alpha\nabla f_i) \circ \mathbf{prox}_{\alpha r}(\mathcal{A}\boldsymbol{z}), \cdots, z_n\}$$

where $I$ denotes the identity mapping. When applying $\mathcal{T}_i$, it is noted that the $i$-th block coordinate in $\boldsymbol{z}$ is updated while the others remain unchanged.

**Proposition 1.** *Prox-DFinito with fixed cyclic sampling order $\pi$ is equivalent to the following fixed-point recursion (see proof in Appendix B.1.)*

$$\boldsymbol{z}^{(k+1)n} = (1-\theta)\boldsymbol{z}^{kn} + \theta\mathcal{T}_\pi\boldsymbol{z}^{kn} \tag{6}$$

*where $\mathcal{T}_\pi = \mathcal{T}_{\pi(n)} \circ \cdots \circ \mathcal{T}_{\pi(1)}$. Furthermore, variable $x^t$ can be recovered by*

$$x^t = \mathbf{prox}_{\alpha r} \circ \mathcal{A}\boldsymbol{z}^t, \quad t = 0, 1, 2, \cdots \tag{7}$$

Similar result also hold for random reshuffling scenario.

**Proposition 2.** *Prox-DFinito with random reshuffling is equivalent to*

$$z^{(k+1)n} = (1-\theta)z^{kn} + \theta\mathcal{T}_{\tau_k}z^{kn} \tag{8}$$

*where $\mathcal{T}_{\tau_k} = \mathcal{T}_{\tau_k(n)} \circ \cdots \circ \mathcal{T}_{\tau_k(1)}$. Furthermore, variable $x^t$ can be recovered by following* (7).

### 2.2 Optimality condition

Assume there exists $x^\star$ that minimizes $F(x) + r(x)$, i.e., $0 \in \nabla F(x^\star) + \partial r(x^\star)$. Then the relation between the minimizer $x^\star$ and the fixed-point $z^\star$ of recursion (6) and (8) can be characterized as:

**Proposition 3.** $x^\star$ *minimizes $F(x)+r(x)$ if and only if there is $z^\star$ so that (proof in Appendix B.2)*

$$z^\star = \mathcal{T}_i z^\star, \quad \forall i \in [n], \tag{9}$$
$$x^\star = \mathbf{prox}_{\alpha r} \circ \mathcal{A}z^\star. \tag{10}$$

**Remark 1.** *If $x^\star$ minimizes $F(x) + r(x)$, it holds from* (9) *and* (10) *that $z_i^\star = (I - \alpha\nabla f_i) \circ \mathbf{prox}_{\alpha r}(\mathcal{A}z^\star) = x^\star - \alpha\nabla f_i(x^\star)$ for any $i \in [n]$.*

### 2.3 An order-specific norm

To gauge the influence of different sampling orders, we now introduce an *order-specific* norm.

**Definition 1.** *Given $z = col\{z_1, \cdots, z_n\} \in \mathbb{R}^{nd}$ and a fixed cyclic order $\pi$, we define*

$$\|z\|_\pi^2 = \sum_{i=1}^n \frac{i}{n}\|z_{\pi(i)}\|^2 = \frac{1}{n}\|z_{\pi(1)}\|^2 + \frac{2}{n}\|z_{\pi(2)}\|^2 + \cdots + \|z_{\pi(n)}\|^2$$

*as the $\pi$-specific norm.*

For two different cyclic orders $\pi$ and $\pi'$, it generally holds that $\|z\|_\pi^2 \neq \|z\|_{\pi'}^2$. Note that the coefficients in $\|z\|_\pi^2$ are delicately designed for technical reasons (see Lemma 1 and its proof in the appendix). The order-specific norm facilitates the performance comparison between two orderings.

## 3 Convergence Analysis

In this section we establish the convergence rate of Prox-DFinito with cyclic sampling and random reshuffling in convex and strongly convex scenarios, respectively.

### 3.1 The convex scenario

We first study the convex scenario under the following assumption:

**Assumption 1** (Convex). *Each function $f_i(x)$ is convex and $L$-smooth.*

It is worth noting that the convergence results on cyclic sampling and random reshuffling for the convex scenario are quite limited except for [22, 33, 5, 18].

**Cyclic sampling and shuffling-once.** We first introduce the following lemma showing that $\mathcal{T}_\pi$ is non-expansive with respect to $\|\cdot\|_\pi$, which is fundamental to the convergence analysis.

**Lemma 1.** *Under Assumption 1, if step-size $0 < \alpha \leq \frac{2}{L}$ and the data is sampled with a fixed cyclic order $\pi$, it holds that (see proof in Appendix C.1)*

$$\|\mathcal{T}_\pi u - \mathcal{T}_\pi v\|_\pi^2 \leq \|u - v\|_\pi^2, \quad \forall u, v \in \mathbb{R}^{nd}. \tag{11}$$

Recall (6) that the sequence $z^{kn}$ is generated through $z^{(k+1)n} = \mathcal{S}_\pi z^{kn}$. Since $\mathcal{S}_\pi = (1-\theta)I + \theta\mathcal{T}_\pi$ and $\mathcal{T}_\pi$ is non-expansive, we can prove the distance $\|z^{(k+1)n} - z^{kn}\|^2$ will converge to 0 sublinearly:

**Lemma 2.** *Under Assumption 1, if step-size $0 < \alpha \leq \frac{2}{L}$ and the data is sampled with a fixed cyclic order $\pi$, it holds for any $k = 0, 1, \cdots$ that (see proof in Appendix*

$$\|z^{(k+1)n} - z^{kn}\|_\pi^2 \leq \frac{\theta}{(k+1)(1-\theta)}\|z^0 - z^\star\|_\pi^2 \tag{12}$$

*where $\theta \in (0, 1)$ is the damping parameter.*

With Lemma 2 and the relation between $x^t$ and $z^t$ in (7), we can establish the convergence rate:

**Theorem 1.** *Under Assumption 1, if step-size $0 < \alpha \leq \frac{2}{L}$ and the data is sampled with a fixed cyclic order $\pi$, it holds that (see proof in Appendix C.3)*

$$\min_{g \in \partial r(x^{kn})} \|\nabla F(x^{kn}) + g\|^2 \leq \frac{CL^2}{(k+1)\theta(1-\theta)} \tag{13}$$

*where $\theta \in (0,1)$ and $C = \left(\frac{2}{\alpha L}\right)^2 \frac{\log(n)+1}{n} \|z^0 - z^\star\|_\pi^2$.*

**Remark 2.** *Inspired by reference [16], one can take expectation over cyclic order $\pi$ in (13) to obtain the convergence rate of Prox-DFinito shuffled once before training begins (with $C = \left(\frac{2}{\alpha L}\right)^2 \frac{(n+1)(\log(n)+1)}{2n^2} \|z^0 - z^\star\|^2$):*

$$\mathbb{E} \min_{g \in \partial r(x^{kn})} \|\nabla F(x^{kn}) + g\|^2 \leq \frac{CL^2}{(k+1)\theta(1-\theta)} \tag{14}$$

**Random reshuffling.** We let $\tau_k$ denote the sampling order used in the $k$-th epoch. Apparently, $\tau_k$ is a uniformly distributed random variable with $n!$ realizations. With the similar analysis technique, we can also establish the convergence rate under random reshuffling in the expectation sense.

**Theorem 2.** *Under Assumption 1, if step-size $0 < \alpha \leq \frac{2}{L}$ and data is sampled with random reshuffling, it holds that (see proof in Appendix D.2)*

$$\mathbb{E} \min_{g \in \partial r(x^{kn})} \|\nabla F(x^{kn}) + g\|^2 \leq \frac{CL^2}{(k+1)\theta(1-\theta)} \tag{15}$$

*where $\theta \in (0,1)$ and $C = \left(\frac{5}{3\alpha L}\right)^2 \frac{1}{n} \|z^0 - z^\star\|^2$.*

Comparing (15) with (13), it is observed that random reshuffling replaces the constant $\|z^0 - z^\star\|_\pi^2$ by $\|z^0 - z^\star\|^2$ and removes the $\log(n)$ term in the upper bound.

### 3.2 The strongly convex scenario

In this subsection, we study the convergence rate of Prox-DFinito under the following assumption:

**Assumption 2** (Strongly Convex). *Each function $f_i(x)$ is $\mu$-strongly convex and $L$-smooth.*

**Theorem 3.** *Under Assumption 2, if step-size $0 < \alpha \leq \frac{2}{\mu+L}$, it holds that (see proof in Appendix E)*

$$(\mathbb{E}) \|x^{kn} - x^\star\|^2 \leq \left(1 - \frac{2\theta\alpha\mu L}{\mu + L}\right)^k C \tag{16}$$

*where $\theta \in (0,1)$ and*

$$C = \begin{cases} \frac{\log(n)+1}{n} \|z^0 - z^\star\|_\pi^2 & \text{with } \pi\text{-order cyclic sampling}, \\ \frac{1}{n} \|z^0 - z^\star\|^2 & \text{with random reshuffling}. \end{cases}$$

**Remark 3.** *Note when $\theta \to 1$, Prox-DFinito actually reaches the best performance, so damping is essentially not necessary in strongly convex scenario.*

### 3.3 Comparison with the existing results

Recalling $\|z\|_\pi^2 = \sum_{i=1}^n \frac{i}{n} \|z_{\pi(i)}\|^2$, it holds that

$$\frac{1}{n} \|z\|^2 \leq \|z\|_\pi^2 \leq \|z\|^2, \quad \forall z, \pi. \tag{17}$$

For a fair comparison with existing works, we consider the *worst case* performance of cyclic sampling by relaxing $\|z^0 - z^\star\|_\pi^2$ to its upper bound $\|z^0 - z^\star\|^2$. Letting $\alpha = \mathcal{O}(1/L)$, $\theta = 1/2$ and assuming $\frac{1}{n} \|z^0 - z^\star\|^2 = \mathcal{O}(1)$, the convergence rates derived in Theorems 1–3 reduce to

$$\text{C-Cyclic} = \tilde{\mathcal{O}}(L^2/k), \qquad \text{C-RR} = \mathcal{O}(L^2/k)$$
$$\text{SC-Cyclic} = \tilde{\mathcal{O}}((1 - 1/\kappa)^k), \qquad \text{SC-RR} = \mathcal{O}((1 - 1/\kappa)^k).$$

where "C" denotes "convex" and "SC" denotes "strongly convex", $\kappa = L/\mu$, and $\tilde{\mathcal{O}}(\cdot)$ hides the $\log(n)$ factor. Note that all rates are in the epoch-wise sense. These rates can be translated into the the gradient complexity (equivalent to sample complexity) of Prox-DFinito to reach an $\epsilon$-accurate solution. The comparison with existing works are listed in Table 1.

**Different metrics.** Except for [5] and our Prox-DFinito algorithm whose convergence analyses are based on the gradient norm in the convex and smooth scenario, results in other references are based on function value metric (i.e., objective error $F(x^{kn}) - F(x^\star)$). The function value metric can imply the gradient norm metric, but not always vice versa. To comapre Prox-DFinito with other established results in the same metric, we have to transform the rates in other references into the gradient norm metric. The comparison is listed in Table 1. When the gradient norm metric is used, we observe that the rates of Prox-DFinito match that with gradient descent, and are state-of-the-art compared to the existing results. However, the rate of Prox-DFinito in terms of the function value is not known yet (this unknown rate may end up being worse than those of the other methods).

For the non-smooth scenario, our metric $\min_{g \in \partial r(x)} \|\nabla F(x) + \partial r(x)\|^2$ may not be bounded by the functional suboptimality $F(x) + r(x) - F(x^\star) - r(x^\star)$, and hence Prox-DFinito results are not comparable with those in [21, 35, 37, 33, 18]. The results listed in Table 1 are all for the smooth scenario of [21, 35, 37, 33, 18], and we use "Support Prox" to indicate whether the results cover the non-smooth scenario or not.

**Assumption scope.** Except for references [18, 35] and Proximal GD algorithm whose convergence analyses are conducted by assuming the average of each function to be $\bar{L}$-smooth (and perhaps $\bar{\mu}$-strongly convex), results in other references are based on a stronger assumption that each summand function to be $L$-smooth (and perhaps $\mu$-strongly convex). Note that $\bar{L}$ can be much smaller than $L$ sometimes. To compare [18, 35] and Proximal GD with other references under the same assumption, we let each $L = \bar{L}$ in Table 1. However, it is worth noting that when each $L_i$ is drastically different from each other and can be evaluated precisely, the results relying on $\bar{L}$ (e.g., [35] and [18]) can be much better than the results established in this work.

**Comparison with GD.** It is observed from Table 1 that Prox-DFinito with cyclic sampling or random reshuffling is *no-worse* than Proximal GD. It is the first *no-worse-than-GD* result, besides the independent and concurrent work [18], that covers both the non-smooth and the convex scenarios for variance-reduction methods under without-replacement sampling orders. The pioneering work DIAG [23] established a similar result only for smooth and strongly-convex problems[2].

**Comparison with RR/CS methods.** Prox-DFinito achieves the nearly state-of-the-art gradient complexity in both convex and strongly convex scenarios (except for the convex and smooth case due to the weaker metric adopted) among known without-replacement stochastic approaches to solving the finite-sum optimization problem (1), see Table 1. In addition, it is worth noting that in Table 1, algorithms of [33, 35, 23] and our Prox-DFinito have an $\mathcal{O}(nd)$ memory requirement while others only need $\mathcal{O}(d)$ memory. In other words, Prox-DFinito is memory-costly in spite of its superior theoretical convergence rate and sample complexity.

**Comparison with uniform-iid-sampling methods.** It is known that uniform-sampling variance-reduction can achieve an $\mathcal{O}(\max\{n, L/\mu\}\log(1/\epsilon))$ sample complexity for strongly convex problems [14, 26, 6] and $\mathcal{O}(L^2/\epsilon)$ (when using metric $\mathbb{E}\|\nabla F(x)\|^2$) for convex problems [26]. In other words, these uniform-sampling methods have sample complexities that are independent of sample size $n$. Our achieved results (and other existing results listed in Table 1 and [18]) for random reshuffling or worst-case cyclic sampling cannot match with uniform-sampling yet. However, this paper establishes that Prox-DFinito with the optimal cyclic order, in the highly data-heterogeneous scenario, can achieve an $\tilde{\mathcal{O}}(L^2/\epsilon)$ sample complexity in the convex scenario, which matches with uniform-sampling up to a $\log(n)$ factor, see the detailed discussion in Sec. 4. To our best knowledge, it is the first result, at least in certain scenarios, that variance reduction under without-replacement sampling orders can match with its uniform-sampling counterpart in terms of their sample complexity upper bound. Nevertheless, it still remains unclear how to close the gap in sample complexity between variance reduction under without-replacement sampling and uniform sampling in the more general settings (i.e., settings other than highly data-heterogeneous scenario).

---

[2]While DIAG is established to outperform gradient descent in [23], we find its convergence rate is still on the same order of GD. Its superiority to GD comes from the constant improvement, not order improvement.

# 4  Optimal Cyclic Sampling Order

Sec.3.3 examines the worst case gradient complexity of Prox-DFinito with cyclic sampling, which is worse than random reshuffling by a factor of $\log(n)$ in both convex and strongly convex scenarios. In this section we examine how Prox-DFinito performs with *optimal* cyclic sampling.

## 4.1  Optimal cyclic sampling

Given sample size $n$, step-size $\alpha$, epoch index $k$, and constants $L$, $\mu$ and $\theta$, it is derived from Theorem 1 that the rate of $\pi$-order cyclic sampling is determined by constant

$$\|\boldsymbol{z}^0 - \boldsymbol{z}^\star\|_\pi^2 = \sum_{i=1}^{n} \frac{i}{n}\|z_{\pi(i)}^0 - z_{\pi(i)}^\star\|^2. \tag{18}$$

We define the corresponding optimal cyclic order as follows.

**Definition 2.** *An optimal cyclic sampling order $\pi^\star$ of Prox-DFinito is defined as*

$$\pi^\star := \arg\min_\pi \{\|\boldsymbol{z}^0 - \boldsymbol{z}^\star\|_\pi^2\}. \tag{19}$$

Such an optimal cyclic order can be identified as follows (see proof in Appendix F).

**Proposition 4.** *The optimal cyclic order for Prox-DFinito is the reverse order of $\{\|z_i^0 - z_i^\star\|^2\}_{i=1}^n$.*

**Remark 4** (IMPORTANCE INDICATOR). *Proposition 4 implies that $\|z_i^0 - z_i^\star\|^2$ can be used as an importance indicator of sample $i$. Recall $z_i^\star = x^\star - \alpha\nabla f_i(x^\star)$ from Remark 1. If $z_i^0$ is initialized as $0$, the importance indicator of sample $i$ reduces to $\|x^\star - \alpha\nabla f_i(x^\star)\|^2$, which is determined by both $x^\star$ and $\nabla f_i(x^\star)$. If $z_i^0$ is initialized close to $x^\star$, we then have $\|z_i^0 - z_i^\star\|^2 \approx \alpha^2\|\nabla f_i(x^\star)\|^2$. In other words, the importance of sample $i$ can be measured by $\|\nabla f_i(x^\star)\|$, which is consistent with the importance indicator in uniform-iid-sampling [41, 40].*

## 4.2  Optimal cyclic sampling can achieve sample-size-independent complexity

Recall from Theorem 1 that the sample complexity of Prox-DFinito with cyclic sampling in the convex scenario is determined by $(\log(n)/n)\|\boldsymbol{z}^0 - \boldsymbol{z}^\star\|_\pi^2$. From (17) we have

$$\frac{1}{n}\|\boldsymbol{z}^0 - \boldsymbol{z}^\star\|^2 \le \|\boldsymbol{z}^0 - \boldsymbol{z}^\star\|_\pi^2 \le \|\boldsymbol{z}^0 - \boldsymbol{z}^\star\|^2, \quad \forall \boldsymbol{z}, \pi. \tag{20}$$

In Sec. 3.3 we considered the worst case performance of cyclic sampling, i.e., we bound $\|\boldsymbol{z}^0 - \boldsymbol{z}^\star\|_\pi^2$ with its upper bound $\|\boldsymbol{z}^0 - \boldsymbol{z}^\star\|^2$. In this section, we will examine the best case performance using the lower bound $\|\boldsymbol{z}^0 - \boldsymbol{z}^\star\|^2/n$, and provide a scenario in which such best case performance is achievable. We assume $\|\boldsymbol{z}^0 - \boldsymbol{z}^\star\|^2/n = \mathcal{O}(1)$ as in previous sections.

**Proposition 5.** *Given fixed constants $n$, $\alpha$, $k$, $\theta$, $L$, and optimal cyclic order $\pi^\star$, if the condition*

$$\rho := \frac{\|\boldsymbol{z}^0 - \boldsymbol{z}^\star\|_{\pi^\star}^2}{\|\boldsymbol{z}^0 - \boldsymbol{z}^\star\|^2} = \mathcal{O}\left(\frac{1}{n}\right) \tag{21}$$

*holds, then Prox-DFinito with optimal cyclic sampling achieves sample complexity $\tilde{\mathcal{O}}(L^2/\epsilon)$.*

The above proposition can be proved by directly substituting (21) into Theorem 1. In the following, we discuss a data-heterogeneous scenario in which relation (21) holds.

**A data-heterogeneous scenario.** To this end, we let $\boldsymbol{x}^\star = \text{col}\{x^\star, \cdots, x^\star\}$ and $\nabla\boldsymbol{f}(x^\star) = \text{col}\{\nabla f_1(x^\star), \cdots, \nabla f_n(x^\star)\}$, it follows from Remark 1 that $\boldsymbol{z}^\star = \boldsymbol{x}^\star - \alpha\nabla\boldsymbol{f}(x^\star)$. If we set $\boldsymbol{z}^0 = 0$ (which is common in the implementation) and $\alpha = 1/L$ (the theoretically suggested step-size), it then holds that $\|z_i^0 - z_i^\star\|^2 = \|x^\star - \nabla f_i(x^\star)/L\|^2$. Next, we assume $\|z_i^0 - z_i^\star\|^2 = \|x^\star - \nabla f_i(x^\star)/L\|^2 = n\beta^{i-1}$ ($0 < \beta < 1$) holds. Under such assumption, the optimal cyclic order will be $\pi^\star = (1, 2, \cdots, n)$. Now we examine $\|\boldsymbol{z}^0 - \boldsymbol{z}^\star\|_{\pi^\star}^2$ and $\|\boldsymbol{z}^0 - \boldsymbol{z}^\star\|^2$:

$$\sum_{i=1}^{n}\|z_i^0 - z_i^\star\|^2 = n\sum_{i=1}^{n}\beta^{i-1} \approx \frac{n}{1-\beta}, \quad \sum_{i=1}^{n}\frac{i}{n}\|z_i^0 - z_i^\star\|^2 = \sum_{i=1}^{n}i\beta^{i-1} \approx \frac{1}{(1-\beta)^2}$$

when $n$ is large, which implies that $\rho = \|\boldsymbol{z}^0 - \boldsymbol{z}^\star\|_{\pi^\star}^2/\|\boldsymbol{z}^0 - \boldsymbol{z}^\star\|^2 = \mathcal{O}(1/n)$ since $\beta$ is a constant independent of $n$. With Proposition 5, we know Prox-DFinito with optimal cyclic sampling can achieve $\tilde{\mathcal{O}}(L^2/\epsilon)$, which is independent of sample size $n$. Note that $\|\nabla f_i(x^\star)\|^2 = n\beta^{i-1}$ implies a data-heterogeneous scenario where $\beta$ can roughly gauge the variety of data samples.

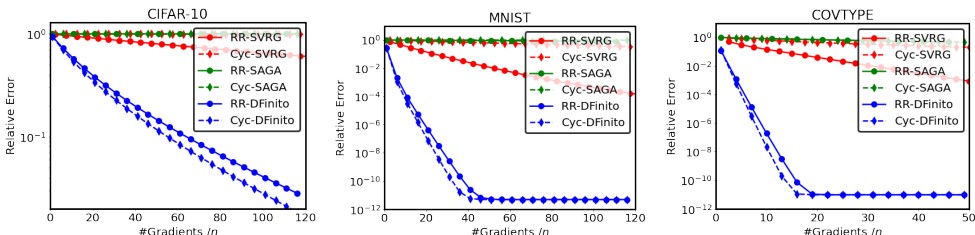

Figure 1: Comparison with SVRG and SAGA under without-replacement sampling orders using theoretical step sizes on Cifar-10 ($\lambda = 0.005$), MNIST ($\lambda = 0.008$), and Covtype ($\lambda = 0.05$). The $y$-axis indicates the relative mean-square error $(\mathbb{E})\|x - x^\star\|^2/\|x^0 - x^\star\|^2$ versus #gradient evaluations/$n$.

## 4.3 Adaptive importance reshuffling

The optimal cyclic order decided by Proposition 4 is not practical since the importance indicator of each sample depends on the unknown $z_i^\star = x^\star - \alpha \nabla f_i(x^\star)$. This problem can be overcome by replacing $z_i^\star$ by its estimate $z_i^{kn}$, which leads to an adaptive importance reshuffling strategy.

---
**Algorithm 2** Adaptive Importance Reshuffling

**Initialize:** $w^0(i) = \|z_i^0 - \bar{z}^0\|^2$ for $i \in [n]$;
**for** epoch $k = 0, 1, 2, \cdots$ **do**
 Reshuffle $[n]$ based on the vector $w^k$;
 Update a Prox-DFinito epoch;
 Update $w^{k+1}$ according to (22);
**end for**

---

We introduce $w \in \mathbb{R}^n$ as an importance indicating vector with each element $w_i$ indicating the importance of sample $i$ and initialized as $w^0(i) = \|z_i^0 - \bar{z}^0\|^2, \ \forall i \in [n]$. In the $k$-th epoch, we draw sample $i$ earlier if $w^k(i)$ is larger. After the $k$-th epoch, $w$ will be updated as

$$w^{k+1}(i) = (1 - \gamma)w^k(i) + \gamma\|z_i^0 - z_i^{(k+1)n}\|^2, \tag{22}$$

where $i \in [n]$ and $\gamma \in (0, 1)$ is a fixed damping parameter. Suppose $z_i^{kn} \to z_i^\star$, the above recursion will guarantee $w^k(i) \to \|z_i^0 - z_i^\star\|^2$. In other words, the order decided by $w^k$ will gradually adapt to the optimal cyclic order as $k$ increases. Since the order decided by importance changes from epoch to epoch, we call this approach *adaptive importance reshuffling* and list it in Algorithm 2. We provide the convergence guarantees of the adaptive importance reshuffling method in Appendix G.

## 5 Numerical Experiments

### 5.1 Comparison with SVRG and SAGA under without-replacement sampling orders

In this experiment, we compare DFinito with SVRG [14] and SAGA [7] under without-replacement sampling (RR, cyclic sampling). We consider a similar setting as in [18, Figure 2], where all step sizes are chosen as the *theoretically* optimal one, see Table 2 in Appendix H. We run experiments for the regularized logistic regression problem, i.e. problem (1) with $f_i(x) = \log\left(1 + \exp(-y_i\langle w_i, x\rangle)\right) + \frac{\lambda}{2}\|x\|^2$ with three widely-used datasets: CIFAR-10 [15], MNIST [8], and COVTYPE [29]. This problem is $L$-smooth and $\mu$-strongly convex with $L = \frac{1}{4n}\lambda_{\max}(W^T W) + \lambda$ and $\mu = \lambda$. From Figure 1, it is observed that DFinito outperforms SVRG and SAGA in terms of gradient complexity under without-replacement sampling orders with their best-known theoretical rates. The comparison with SVRG and SAGA with the *practically* optimal step sizes is in Appendix J.

### 5.2 DFinito with cyclic sampling

**Justification of the optimal cyclic sampling order.** To justify the optimal cyclic sampling order $\pi^\star$ suggested in Proposition 4, we test DFinito with eight arbitrarily-selected cyclic orders, and compare them with the optimal cyclic ordering $\pi^\star$ as well as the adaptive importance reshuffling method (Algorithm 2). To make the comparison distinguishable, we construct a least square problem with heterogeneous data samples with $n = 200$, $d = 50$, $L = 100$, $\mu = 10^{-2}$ (see Appendix I for the constructed problem). The constructed problem is with $\rho = \|z^0 - z^\star\|_{\pi^*}^2/\|z^0 - z^\star\|_2^2 = 0.006$ when $z_i^0 = 0$, $x^0 = 0$, and $\alpha = \frac{1}{3L}$, which is close to $1/n = 0.005$. In the left plot in Fig. 2, it is observed that the optimal cyclic sampling achieves the fastest convergence rate. Furthermore, the adaptive

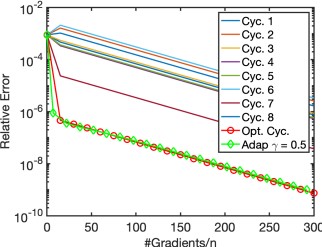 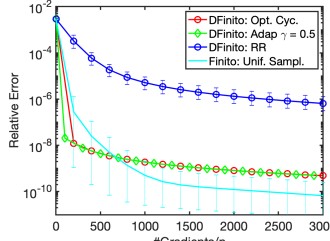

Figure 2: Left: Performance of DFinito under different sampling orders. Error metric: $\|x - x^\star\|^2 / \|x^0 - x^\star\|^2$. Right: Comparison of DFinito under $\pi^\star$-cyclic sampling and uniform sampling in a highly heterogeneous scenario. Error metric: $\|\nabla F(x)\|^2 / \|\nabla F(x^0)\|^2$.

shuffling method can match with the optimal cyclic ordering. These observations are consistent with our theoretical results derived in Sec. 4.2 and 4.3.

**Optimal cyclic sampling can achieve sample-size-independent complexity.** It is established in [26] that Finito with uniform-iid-sampling can achieve $n$-independent gradient complexity with $\alpha = \frac{n}{8L}$. In this experiment, we compare DFinito ($\alpha = \frac{2}{L}$) with Finito under uniform sampling (8 runs, $\alpha = \frac{n}{8L}$) in a convex and highly heterogeneous scenario ($\rho = \mathcal{O}(\frac{1}{n})$). The constructed problem is with $n = 500$, $d = 20$, $L = 0.3$, $\theta = 0.5$ and $\|z_i^0 - z_i^\star\| = 10000 * 0.1^{i-1}$, $1 \leq i \leq n$ (see detailed initialization in Appendix J). We also depict DFinito with random-reshuffling (8 runs) as another baseline. In the right plot of Figure 2, it is observed that the convergence curve of DFinito with $\pi^\star$-cyclic sampling matches with Finito with uniform sampling. This implies DFinito can achieve the same $n$-independent gradient complexity as Finito with uniform sampling.

### 5.3 More experiments

We conduct more experiments in Appendix J. First, we compare DFinito with GD/SGD to justify its empirical superiority to these methods. Second, we validate how different data heterogeneity will influence optimal cyclic sampling. Third, we examine the performance of SVRG, SAGA, and DFinito under without/with-replacement sampling using grid-search (not theoretical) step sizes.

## 6 Conclusion and Discussion

This paper develops Prox-DFinito and analyzes its convergence rate under without-replacement sampling in both convex and strongly convex scenarios. Our derived rates are state-of-the-art compared to existing results. In particular, this paper derives the best-case convergence rate for Prox-DFinito with cyclic sampling, which can be sample-size-independent in the highly data-heterogeneous scenario. A future direction is to close the gap in gradient complexity between variance reduction under without-replacement and uniform-iid-sampling in the more general setting.

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
