# Appendix

## A  Efficient Implementation of Prox-Finito

---
**Algorithm 3** Prox-Finito: Efficient Implementation

---
**Input:** $\bar{z}^0 = \frac{1}{n} \sum_{i=1}^{n} z_i^0$, step-size $\alpha$, and $\theta \in (0,1)$;
 **for** epoch $k = 0, 1, 2, \cdots$ **do**
  **for** iteration $t = kn+1, kn+2, \cdots, (k+1)n$ **do**
   $x^{t-1} = \mathbf{prox}_{\alpha r}(\bar{z}^{t-1})$;
   Pick $i_t$ with some rule;
   Compute $d_{i_t}^t = x^{t-1} - \alpha \nabla f_i(x^{t-1}) - z_{i_t}^{t-1}$;
   Update $\bar{z}^t = \bar{z}^{t-1} + d_{i_t}^t/n$;
   Update $z_{i_t}^t = z_{i_t}^{t-1} + \theta d_{i_t}^t$ and delete $d_{i_t}^t$;
  **end for**
  $\bar{z}^{(k+1)n} \leftarrow (1-\theta)\bar{z}^{kn} + \theta \bar{z}^{(k+1)n}$;
 **end for**

---

## B  Operator's Form

### B.1  Proof of Proposition 1

*Proof.* In fact, it suffices to notice that

$$z^{kn+\ell} = \begin{cases} \mathcal{T}_{\pi(\ell)} z^{kn+\ell-1} & \text{if } \ell \in [n-1] \\ (1-\theta)z^{kn} + \mathcal{T}_{\pi(n)} z^{kn+n-1} & \text{if } \ell = n, \end{cases}$$

and the $x$-update in (7) directly follows (4b). $\qquad\square$

### B.2  Proof of Proposition 3

*Proof.* With definition (2), we can reach the following important relation:

$$x = \mathbf{prox}_{\alpha r}(y) \iff 0 \in \alpha \partial r(x) + x - y. \tag{23}$$

**Sufficiency.** Assuming $x^\star$ minimizes $F(z) + r(x)$, it holds that $0 \in \nabla F(x^\star) + \partial r(x^\star)$. Let $z_i^\star = (I - \alpha \nabla f_i)(x^\star)$ and $\mathbf{z}^\star = \mathrm{col}\{z_1^\star, \ldots, z_n^\star\}$, we now prove $\mathbf{z}^\star$ satisfies (9) and (10).

Note $\mathcal{A}\mathbf{z}^\star = \frac{1}{n} \sum_{i=1}^{n} (I - \alpha \nabla f_i)(x^\star) = x^\star - \alpha \nabla F(x^\star)$ and $0 \in x^\star - (x^\star - \alpha \nabla F(x^\star)) + \alpha \partial r(x^\star)$, it holds that

$$x^\star = \mathbf{prox}_{\alpha r}(x^\star - \alpha \nabla F(x^\star)) = \mathbf{prox}_{\alpha r}(\mathcal{A}\mathbf{z}^\star) \tag{24}$$

and hence

$$(I - \alpha \nabla f_i) \circ \mathbf{prox}_{\alpha r}(\mathcal{A}\mathbf{z}^\star) = (I - \alpha \nabla f_i)(x^\star) = z_i^\star, \quad \forall i \in [n]. \tag{25}$$

Therefore, $\mathbf{z}^\star$ satisfies (9) and (10).

**Necessity.** Assuming $\mathbf{z}^\star = \mathcal{T}_i \mathbf{z}^\star$, $\forall i \in [n]$, we have $z_i^\star = (I - \alpha \nabla f_i) \circ \mathbf{prox}_{\alpha r}(\mathcal{A}\mathbf{z}^\star)$. By averaging all $z_i^\star$, we have

$$\mathcal{A}\mathbf{z}^\star = (I - \alpha \nabla F) \circ \mathbf{prox}_{\alpha r}(\mathcal{A}\mathbf{z}^\star). \tag{26}$$

Let $x^\star = \mathbf{prox}_{\alpha r}(\mathcal{A}\mathbf{z}^\star)$ and apply $\mathbf{prox}_{\alpha r}$ to (26), we reach

$$x^\star = \mathbf{prox}_{\alpha r}(x^\star - \alpha \nabla F(x^\star)), \tag{27}$$

which indicates $0 \in \alpha \partial r(x^\star) + x^\star - (x^\star - \alpha F(x^\star)) \iff 0 \in \nabla F(x^\star) + \partial r(x^\star)$, i.e. $x^\star$ is a minimizer. $\qquad\square$

## C  Cyclic–Convex

### C.1  Proof of Lemma 1

*Proof.* Without loss of generality, we only prove the case in which $\pi = (1, 2, \ldots, n)$ where $\mathcal{T}_\pi = \mathcal{T}_n \circ \cdots \circ \mathcal{T}_2 \circ \mathcal{T}_1$.

To ease the notation, for $\boldsymbol{z} \in \mathbb{R}^{nd}$, we define $h_i$-norm as

$$\|\boldsymbol{z}\|_{h_i}^2 = \frac{1}{n} \sum_{j=1}^{n} \left(\mathrm{mod}_n(j - i - 1) + 1\right) \|z_j\|^2. \tag{28}$$

Note $\|\boldsymbol{z}\|_{h_n}^2 = \|\boldsymbol{z}\|_{h_0}^2 = \|\boldsymbol{z}\|_\pi^2$ when $\pi = (1, 2, \ldots, n)$.

To begin with, we introduce the non-expansiveness of operator $(I - \alpha\nabla f_i) \circ \mathbf{prox}_{\alpha r}$, i.e.

$$\|(I - \alpha\nabla f_i) \circ \mathbf{prox}_{\alpha r}(x) - (I - \alpha\nabla f_i) \circ \mathbf{prox}_{\alpha r}(y)\|^2 \leq \|x - y\|^2, \ \forall\, x, y \in \mathbb{R}^d \text{ and } i \in [n]. \tag{29}$$

Note that $\mathbf{prox}_{\alpha r}$ is non-expansive by itself; see [30, 13]. $I - \alpha\nabla f_i$ is non-expansive because

$$\begin{aligned}
&\|x - \alpha\nabla f_i(x) - y + \alpha\nabla f_i(y)\|^2 \\
&= \|x - y\|^2 - 2\alpha\langle x - y, \nabla f_i(x) - \nabla f_i(y)\rangle + \alpha^2\|\nabla f_i(x) - \nabla f_i(y)\|^2 \\
&\leq \|x - y\|^2 - \left(\frac{2\alpha}{L} - \alpha^2\right)\|\nabla f_i(x) - \nabla f_i(y)\|^2 \\
&\leq \|x - y\|^2 \quad \forall x \in \mathbb{R}^d, y \in \mathbb{R}^d
\end{aligned}$$

where the last inequality holds when $\alpha \leq \frac{2}{L}$. Therefore, the non-expansiveness of $I - \alpha\nabla f_i$ and $\mathbf{prox}_{\alpha r}$ imply that the composition $(I - \alpha\nabla f_i) \circ \mathbf{prox}_{\alpha r}$ is also non-expansive.

We then check the operator $\mathcal{T}_i$. Suppose $\boldsymbol{u} \in \mathbb{R}^{nd}$ and $\boldsymbol{v} \in \mathbb{R}^{nd}$,

$$\begin{aligned}
\|\mathcal{T}_i\boldsymbol{u} - \mathcal{T}_i\boldsymbol{v}\|_{h_i}^2 &= \frac{1}{n} \sum_{j \neq i} \left(\mathrm{mod}_n(j - i - 1) + 1\right)\|u_j - v_j\|^2 \\
&\quad + \|(I - \alpha\nabla f_i) \circ \mathbf{prox}_{\alpha r}(\mathcal{A}\boldsymbol{u}) - (I - \alpha\nabla f_i) \circ \mathbf{prox}_{\alpha r}(\mathcal{A}\boldsymbol{v})\|^2 \\
&\overset{(a)}{\leq} \frac{1}{n} \sum_{j \neq i} \left(\mathrm{mod}_n(j - i - 1) + 1\right)\|u_j - v_j\|^2 + \|\mathcal{A}\boldsymbol{u} - \mathcal{A}\boldsymbol{v}\|^2 \\
&\overset{(b)}{\leq} \frac{1}{n} \sum_{j \neq i} \left(\mathrm{mod}_n(j - i - 1) + 1\right)\|u_j - v_j\|^2 + \frac{1}{n} \sum_{j=1}^{n} \|u_j - v_j\|^2 \\
&= \frac{1}{n} \sum_{j=1}^{n} \left(\mathrm{mod}_n(j - i) + 1\right)\|u_j - v_j\|^2 = \|\boldsymbol{u} - \boldsymbol{v}\|_{h_{i-1}}^2.
\end{aligned} \tag{30}$$

In the above inequalities, the inequality (a) holds due to (29) and (b) holds because

$$\|\mathcal{A}\boldsymbol{u} - \mathcal{A}\boldsymbol{v}\|^2 = \|\frac{1}{n} \sum_{i=1}^{n} (u_i - v_i)\|^2 \leq \frac{1}{n} \sum_{i=1}^{n} \|u_i - v_i\|^2. \tag{31}$$

With inequality (30), we have that

$$\begin{aligned}
\|\mathcal{T}_\pi\boldsymbol{u} - \mathcal{T}_\pi\boldsymbol{v}\|_\pi^2 &= \|\mathcal{T}_n\mathcal{T}_{n-1}\cdots\mathcal{T}_1\boldsymbol{u} - \mathcal{T}_n\mathcal{T}_{n-1}\cdots\mathcal{T}_1\boldsymbol{v}\|_{h_n}^2 \\
&\leq \|\mathcal{T}_{n-1}\cdots\mathcal{T}_1\boldsymbol{u} - \mathcal{T}_{n-1}\cdots\mathcal{T}_1\boldsymbol{v}\|_{h_{n-1}}^2 \\
&\leq \|\mathcal{T}_{n-2}\cdots\mathcal{T}_1\boldsymbol{u} - \mathcal{T}_{n-2}\cdots\mathcal{T}_1\boldsymbol{v}\|_{h_{n-2}}^2 \\
&\leq \cdots \\
&\leq \|\mathcal{T}_1\boldsymbol{u} - \mathcal{T}_1\boldsymbol{v}\|_{h_1}^2 \\
&\leq \|\boldsymbol{u} - \boldsymbol{v}\|_{h_0}^2 = \|\boldsymbol{u} - \boldsymbol{v}\|_\pi^2.
\end{aligned} \tag{32}$$

$\square$

## C.2 Proof of Lemma 2

*Proof.* We define $\mathcal{S}_\pi = (1 - \theta)I + \theta\mathcal{T}_\pi$ to ease the notation. Then $z^{(k+1)n} = \mathcal{S}_\pi z^{kn}$ by Proposition 1. To prove Lemma 2, notice that $\forall\ k = 1, 2, \cdots$

$$
\begin{aligned}
\|z^{(k+1)n} - z^{kn}\|_\pi^2 &= \|\mathcal{S}_\pi z^{kn} - \mathcal{S}_\pi z^{(k-1)n}\|_\pi^2 \\
&\leq (1 - \theta)\|z^{kn} - z^{(k-1)n}\|_\pi^2 + \theta\|\mathcal{T}_\pi z^{kn} - \mathcal{T}_\pi z^{(k-1)n}\|_\pi^2 \\
&\overset{(11)}{\leq} \|z^{kn} - z^{(k-1)n}\|_\pi^2
\end{aligned}
\tag{33}
$$

The above relation implies that $\|z^{(k+1)n} - z^{kn}\|_\pi^2$ is non-increasing. Next,

$$
\begin{aligned}
\|z^{(k+1)n} - z^\star\|_\pi^2 &= \|(1 - \theta)z^{kn} + \theta\mathcal{T}_\pi(z^{kn}) - z^\star\|_\pi^2 \\
&= (1 - \theta)\|z^{kn} - z^\star\|_\pi^2 + \theta\|\mathcal{T}_\pi(z^{kn}) - z^\star\|_\pi^2 - \theta(1 - \theta)\|z^{kn} - \mathcal{T}(z^{kn})\|_\pi^2 \\
&\overset{(c)}{\leq} \|z^{kn} - z^\star\|_\pi^2 - \theta(1 - \theta)\|z^{kn} - \mathcal{T}_\pi(z^{kn})\|_\pi^2 \\
&= \|z^{kn} - z^\star\|_\pi^2 - \frac{1 - \theta}{\theta}\|z^{kn} - \mathcal{S}_\pi(z^{kn})\|_\pi^2 \\
&= \|z^{kn} - z^\star\|_\pi^2 - \frac{1 - \theta}{\theta}\|z^{kn} - z^{(k+1)n}\|_\pi^2.
\end{aligned}
\tag{34}
$$

where equality (c) holds because Proposition 3 implies $\mathcal{T}_\pi z^\star = z^\star$.

Summing the above inequality from $0$ to $k$ we have

$$
\|z^{(k+1)n} - z^\star\|_\pi^2 \leq \|z^0 - z^\star\|_\pi^2 - \frac{1 - \theta}{\theta}\sum_{\ell=0}^{k}\|z^{\ell n} - z^{(\ell+1)n}\|_\pi^2.
\tag{35}
$$

Since $\|z^{(k+1)n} - z^{kn}\|_\pi^2$ is non-increasing, we reach the conclusion. $\qquad\square$

## C.3 Proof of Theorem 1

*Proof.* Since $z_{\pi(j)}^{kn+j-1} = z_{\pi(j)}^{kn}$ for $1 \leq j \leq n$, it holds that

$$
\begin{aligned}
\bar{z}^{(k+1)n} &= (1 - \theta)\bar{z}^{kn} + \theta\left(\bar{z}^{kn} + \sum_{j=1}^{n}\frac{1}{n}\left((I - \alpha\nabla f_{\pi(j)})(x^{kn+j-1}) - z_{\pi(j)}^{kn+j-1}\right)\right) \\
&= (1 - \theta)\bar{z}^{kn} + \theta\left(\bar{z}^{kn} + \sum_{j=1}^{n}\frac{1}{n}\left((I - \alpha\nabla f_{\pi(j)})(x^{kn+j-1}) - z_{\pi(j)}^{kn}\right)\right) \\
&= (1 - \theta)\bar{z}^{kn} + \theta\sum_{j=1}^{n}\frac{1}{n}(I - \alpha\nabla f_{\pi(j)})(x^{kn+j-1}),
\end{aligned}
\tag{36}
$$

which further implies that

$$
\frac{1}{n}\sum_{j=1}^{n}\nabla f_{\pi(j)}(x^{kn+j-1}) = \frac{1}{\theta\alpha}(\bar{z}^{kn} - \bar{z}^{(k+1)n}) + \frac{1}{n\alpha}\sum_{j=1}^{n}(x^{kn+j-1} - \bar{z}^{kn}).
\tag{37}
$$

As a result, we achieve

$$
\nabla F(x^{kn}) = \frac{1}{n}\sum_{j=1}^{n}\nabla f_{\pi(j)}(x^{kn})
$$

$$
=\frac{1}{n}\sum_{j=1}^{n}\left(\nabla f_{\pi(j)}(x^{kn}) - \nabla f_{\pi(j)}(x^{kn+j-1})\right) + \frac{1}{n}\sum_{j=1}^{n}\nabla f_{\pi(j)}(x^{kn+j-1})
$$

$$
=\frac{1}{n}\sum_{j=1}^{n}\left(\nabla f_{\pi(j)}(x^{kn}) - \nabla f_{\pi(j)}(x^{kn+j-1})\right) + \frac{1}{\theta\alpha}(\bar{z}^{kn} - \bar{z}^{(k+1)n}) + \frac{1}{n\alpha}\sum_{j=1}^{n}(x^{kn+j-1} - \bar{z}^{kn})
$$

$$
=\frac{1}{n\alpha}\sum_{j=1}^{n}\left((I - \alpha\nabla f_{\pi(j)})(x^{kn+j-1}) - (I - \alpha\nabla f_{\pi(j)})(x^{kn})\right)
$$

$$
+ \frac{1}{\theta\alpha}(\bar{z}^{kn} - \bar{z}^{(k+1)n}) + \frac{1}{\alpha}(x^{kn} - \bar{z}^{kn}). \tag{38}
$$

Notice that

$$
x^{kn} = \mathbf{prox}_{\alpha r}(\bar{z}^{kn})
$$
$$
\Longleftrightarrow 0 \in \alpha\,\partial r(x^{kn}) + (x^{kn} - \bar{z}^{kn}) \tag{39}
$$
$$
\Longleftrightarrow \frac{1}{\alpha}(\bar{z}^{kn} - x^{kn}) \triangleq \tilde{\nabla}r(x^{kn}) \in \partial r(x^{kn}),
$$

relation (38) can be rewritten as

$$
\nabla F(x^{kn}) + \tilde{\nabla}r(x^{kn})
$$
$$
=\frac{1}{n\alpha}\sum_{j=1}^{n}\left((I - \alpha\nabla f_{\pi(j)})(x^{kn+j-1}) - (I - \alpha\nabla f_{\pi(j)})(x^{kn})\right) + \frac{1}{\theta\alpha}(\bar{z}^{kn} - \bar{z}^{(k+1)n}). \tag{40}
$$

Next we bound the two terms on the right hand side of (40) by $\|\boldsymbol{z}^{(k+1)n} - \boldsymbol{z}^{kn}\|_{\pi}^{2}$. For the second term, it is easy to see

$$
\|\frac{1}{\theta\alpha}(\bar{z}^{kn} - \bar{z}^{(k+1)n})\|^2 = \frac{1}{n^2\theta^2\alpha^2}\|\sum_{j=1}^{n} z_{\pi(j)}^{kn} - z_{\pi(j)}^{(k+1)n}\|^2
$$
$$
\overset{(d)}{\le} \frac{1}{n^2\theta^2\alpha^2}(\sum_{j=1}^{n}\frac{n}{j})(\sum_{j=1}^{n}\frac{j}{n}\|z_{\pi(j)}^{kn} - z_{\pi(j)}^{(k+1)n}\|^2) \tag{41}
$$
$$
\overset{(e)}{\le} \frac{\log(n)+1}{n\theta^2\alpha^2}\|\boldsymbol{z}^{kn} - \boldsymbol{z}^{(k+1)n}\|_{\pi}^2,
$$

where inequality (d) is due to Cauchy's inequality $(\sum_{j=1}^{n} a_j)^2 \le \sum_{j=1}^{n}\frac{1}{\beta_j}\sum_{j=1}^{n}\beta_j a_j^2$ with $\beta_j > 0, \forall j \in [n]$ and inequality (e) holds because $\sum_{j=1}^{n}\frac{1}{j} \le \log(n)+1$.

For the first term, we first note for $2 \le j \le n$,

$$
z_{\pi(\ell)}^{kn+j-1} = \begin{cases} z_{\pi(\ell)}^{kn} + \frac{1}{\theta}(z_{\pi(\ell)}^{(k+1)n} - z_{\pi(\ell)}^{kn}), & 1 \le \ell \le j-1; \\ z_{\pi(\ell)}^{kn}, & \ell > j-1. \end{cases} \tag{42}
$$

By (29), we have

$$\begin{aligned}
&\|(I - \alpha\nabla f_{\pi(j)})(x^{kn+j-1}) - (I - \alpha\nabla f_{\pi(j)})(x^{kn})\|^2 \\
=&\|(I - \alpha\nabla f_{\pi(j)}) \circ \mathbf{prox}_{\alpha r}(\bar{z}^{kn+j-1}) - (I - \alpha\nabla f_{\pi(j)}) \circ \mathbf{prox}_{\alpha r}(\bar{z}^{kn})\|^2 \\
\leq&\|\bar{z}^{kn+j-1} - \bar{z}^{kn}\|^2 = \|\frac{1}{n}\sum_{\ell=1}^{n}(z_{\pi(\ell)}^{kn+j-1} - z_{\pi(\ell)}^{kn})\|^2 \\
=&\frac{1}{n^2\theta^2}\|\sum_{\ell=1}^{j-1}(z_{\pi(\ell)}^{(k+1)n} - z_{\pi(\ell)}^{kn})\|^2 \leq \frac{1}{n^2\theta^2}\sum_{\ell=1}^{j-1}\frac{n}{\ell}\sum_{\ell=1}^{j-1}\frac{\ell}{n}\|z_{\pi(\ell)}^{(k+1)n} - z_{\pi(\ell)}^{kn}\|^2 \\
\leq&\frac{\log(n)+1}{n\theta^2}\|\boldsymbol{z}^{(k+1)n} - \boldsymbol{z}^{kn}\|_\pi^2.
\end{aligned} \tag{43}$$

In the last inequality, we used the algebraic inequality that $\sum_{\ell=1}^{n}\frac{1}{\ell} \leq \log(n) + 1$. Therefore we have

$$\begin{aligned}
&\|\frac{1}{n\alpha}\sum_{j=1}^{n}\left((I - \alpha\nabla f_{\pi(j)})(x^{kn+j-1}) - (I - \alpha\nabla f_{\pi(j)})(x^{kn})\right)\|^2 \\
=&\frac{1}{n^2\alpha^2}\|\sum_{j=2}^{n}\left((I - \alpha\nabla f_{\pi(j)})(x^{kn+j-1}) - (I - \alpha\nabla f_{\pi(j)})(x^{kn})\right)\|^2 \\
\leq&\frac{1}{n^2\alpha^2}(n-1)^2\frac{\log(n)+1}{n\theta^2}\|\boldsymbol{z}^{(k+1)n} - \boldsymbol{z}^{kn}\|_\pi^2 \\
\leq&\frac{\log(n)+1}{n\theta^2\alpha^2}\|\boldsymbol{z}^{kn} - \boldsymbol{z}^{(k+1)n}\|_\pi^2.
\end{aligned} \tag{44}$$

Combining (41) and (44), we immediately obtain

$$\begin{aligned}
&\min_{g\in\partial r(x^{kn})}\|\nabla F(x^{kn}) + g\|^2 \leq \|\nabla F(x^{kn}) + \tilde{\nabla}r(x^{kn})\|^2 \\
\leq&2(\|\frac{1}{n\alpha}\sum_{j=1}^{n}\left((I - \alpha\nabla f_{\pi(j)})(x^{kn+j-1}) - (I - \alpha\nabla f_{\pi(j)})(x^{kn})\right)\|^2 + \|\frac{1}{\theta\alpha}(\bar{z}^{kn} - \bar{z}^{(k+1)n})\|^2) \\
\leq&2(\frac{\log(n)+1}{n\theta^2\alpha^2}\|\boldsymbol{z}^{kn} - \boldsymbol{z}^{(k+1)n}\|_\pi^2 + \frac{\log(n)+1}{n\theta^2\alpha^2}\|\boldsymbol{z}^{kn} - \boldsymbol{z}^{(k+1)n}\|_\pi^2) \\
=&\left(\frac{2}{\alpha L}\right)^2\frac{(\log(n)+1)L^2}{n\theta^2}\|\boldsymbol{z}^{kn} - \boldsymbol{z}^{(k+1)n}\|_\pi^2 \\
\leq&\left(\frac{2}{\alpha L}\right)^2\frac{L^2}{\theta(1-\theta)(k+1)}\frac{\log(n)+1}{n}\|\boldsymbol{z}^0 - \boldsymbol{z}^\star\|_\pi^2.
\end{aligned} \tag{45}$$

$\square$

# D    RR–Convex

## D.1    Non-expansiveness Lemma for RR

While replacing order-specific norm with standard $\ell_2$ norm. the following lemma establishes that $\mathcal{T}_{\tau_k}$ is non-expansive in expectation.

**Lemma 3.** *Under Assumption 1, if step-size $0 < \alpha \leq \frac{2}{L}$ and data is sampled with random reshuffling, it holds that*

$$\mathbb{E}_{\tau_k}\|\mathcal{T}_{\tau_k}\boldsymbol{u} - \mathcal{T}_{\tau_k}\boldsymbol{v}\|^2 \leq \|\boldsymbol{u} - \boldsymbol{v}\|^2. \tag{46}$$

It is worth noting that inequality (46) holds for $\ell_2$-norm rather than the order-specific norm due to the randomness brought by random reshuffling.

*Proof.* Given any vector $h = [h(1), h(2), \cdots, h(n)]^T \in \mathbb{R}^n$ with positive elements where $h_i$ denotes the $i$-th element of $h$, define $h$-norm as follows

$$\|z\|_h^2 = \sum_{i=1}^n h(i)\|z_i\|^2 \tag{47}$$

for any $z = \text{col}\{z_1, z_2, \cdots, z_n\} \in \mathbb{R}^{nd}$. Following arguments in (30), it holds that

$$\|\mathcal{T}_i u - \mathcal{T}_i v\|_h^2 \le \|u - v\|_{h'}^2 \tag{48}$$

where $h' = h + \frac{1}{n}h(i)\mathbb{1}_n - h(i)e_i$ and $e_i$ is the $i$-th unit vector. Define

$$M_i := I + \frac{1}{n}m_i e_i^T \in \mathbb{R}^{n\times n} \tag{49}$$

where $m_i = \mathbb{1}_n - ne_i$, then we can summarize the above conclusion as follows.

**Lemma 4.** *Given $h \in \mathbb{R}^n$ with positive elements and its corresponding $h$-norm, under Assumption 1, if step-size $0 < \alpha \le \frac{2}{L}$, it holds that*

$$\|\mathcal{T}_i u - \mathcal{T}_i v\|_h^2 \le \|u - v\|_{M_i h}^2 \quad \forall u, v \in \mathbb{R}^{nd}. \tag{50}$$

Therefore, with Lemma 4, we have that

$$
\begin{aligned}
\|\mathcal{T}_\tau u - \mathcal{T}_\tau v\|^2 &= \|\mathcal{T}_{\tau(n)}\mathcal{T}_{\tau(n-1)}\cdots\mathcal{T}_{\tau(1)}u - \mathcal{T}_{\tau(n)}\mathcal{T}_{\tau(n-1)}\cdots\mathcal{T}_{\tau(1)}v\|_{\mathbb{1}_n}^2 \\
&\le \|\mathcal{T}_{\tau(n-1)}\cdots\mathcal{T}_{\tau(1)}u - \mathcal{T}_{\tau(n-1)}\cdots\mathcal{T}_{\tau(1)}v\|_{M_{\tau(n)}\mathbb{1}_n}^2 \\
&\le \|\mathcal{T}_{\tau(n-2)}\cdots\mathcal{T}_{\tau(1)}u - \mathcal{T}_{\tau(n-2)}\cdots\mathcal{T}_{\tau(1)}v\|_{M_{\tau(n-1)}M_{\tau(n)}\mathbb{1}_n}^2 \\
&\le \cdots \\
&\le \|\mathcal{T}_{\tau(1)}u - \mathcal{T}_{\tau(1)}v\|_{M_{\tau(2)}\cdots M_{\tau(n)}\mathbb{1}_n}^2 \\
&\le \|u - v\|_{M_{\tau(1)}\cdots M_{\tau(n)}\mathbb{1}_n}^2.
\end{aligned}
\tag{51}
$$

With the above relation, if we can prove

$$\mathbb{E}_\tau M_{\tau(1)}\cdots M_{\tau(n)}\mathbb{1}_n = \mathbb{1}_n, \tag{52}$$

then we can complete the proof by

$$\mathbb{E}_\tau \|\mathcal{T}_\tau u - \mathcal{T}_\tau v\|^2 \le \mathbb{E}_\tau \|u - v\|_{M_{\tau(1)}\cdots M_{\tau(n)}\mathbb{1}_n}^2 = \|u - v\|_{\mathbb{E}\,M_{\tau(1)}\cdots M_{\tau(n)}\mathbb{1}_n}^2 = \|u - v\|^2. \tag{53}$$

To prove (52), we notice that $e_i^T m_j = 1, \forall i \ne j$ which leads to $m_{\tau(j_1)}e_{\tau(j_1)}^T m_{\tau(j_2)}e_{\tau(j_2)}^T \cdots m_{\tau(j_t)}e_{\tau(j_t)}^T = m_{\tau(j_1)}e_{\tau(j_t)}^T, \forall j_1 < j_2 < \cdots < j_t$. This fact further implies that

$$
\begin{aligned}
M_{\tau(1)}\cdots M_{\tau(n)} &= (I + \frac{1}{n}m_{\tau(1)}e_{\tau(1)}^T)\cdots(I + \frac{1}{n}m_{\tau(n)}e_{\tau(n)}^T) \\
&= I + \frac{1}{n}\sum_{i=1}^n m_i e_i^T + \sum_{t=2}^n \sum_{j_1 < \cdots < j_t} \frac{1}{n^t}m_{\tau(j_1)}e_{\tau(j_1)}^T \cdots m_{\tau(j_t)}e_{\tau(j_t)}^T \\
&= I + \frac{1}{n}\sum_{i=1}^n m_i e_i^T + \sum_{t=2}^n \sum_{i+t-1\le j} \binom{j-i-1}{t-2}\frac{1}{n^t}m_{\tau(i)}e_{\tau(j)}^T \\
&= I + \frac{1}{n}\sum_{i=1}^n m_i e_i^T + \sum_{i<j}\sum_{t=2}^{j-i+1} \binom{j-i-1}{t-2}\frac{1}{n^t}m_{\tau(i)}e_{\tau(j)}^T \\
&= I + \frac{1}{n}\sum_{i=1}^n m_i e_i^T + \sum_{i<j} m_{\tau(i)}e_{\tau(j)}^T \frac{1}{n^2}(1+\frac{1}{n})^{j-i-1}.
\end{aligned}
\tag{54}
$$

It is easy to verify $\sum_{i=1}^{n} m_i e_i^T \mathbb{1}_n = 0$ and

$$
\begin{aligned}
\mathbb{E}_\tau \, m_{\tau(i)} e_{\tau(j)}^T \mathbb{1}_n &= \frac{1}{n(n-1)} \sum_{i \neq j} m_i e_j^T \mathbb{1}_n \\
&= \frac{1}{n(n-1)} \left( \left(\sum_{i=1}^{n} m_i\right) \left(\sum_{j=1}^{n} e_j\right)^T - \sum_{i=1}^{n} m_i e_i^T \right) \mathbb{1}_n = 0.
\end{aligned}
\tag{55}
$$

We can prove (52) by combining (54) and (55). $\qquad\square$

### D.2  Proof of Theorem 2

*Proof.* In fact, with similar arguments of Appendix C.2 and noting $\mathcal{T}_\tau z^\star = z^\star$ for any realization of $\tau$, we can achieve

**Lemma 5.** *Under Assumption 1, if step-size $0 < \alpha \leq \frac{2}{L}$ and the data is sampled with random reshuffling, it holds for any $k = 0, 1, \cdots$ that*

$$
\mathbb{E} \, \|z^{(k+1)n} - z^{kn}\|^2 \leq \frac{\theta}{(k+1)(1-\theta)} \|z^0 - z^\star\|^2.
\tag{56}
$$

Based on Lemma (5), we are now able to prove Theorem 2. By arguments similar to Appendix C.3, $\exists \tilde{\nabla} r(x^{kn}) = \frac{1}{\alpha}(\bar{z}^{kn} - x^{kn}) \in \partial r(x^{kn})$ such that

$$
\begin{aligned}
&\nabla F(x^{kn}) + \tilde{\nabla} r(x^{kn}) \\
&= \frac{1}{n\alpha} \sum_{j=1}^{n} \left( (I - \alpha \nabla f_{\tau_k(j)})(x^{kn+j-1}) - (I - \alpha \nabla f_{\tau_k(j)})(x^{kn}) \right) + \frac{1}{\theta\alpha}(\bar{z}^{kn} - \bar{z}^{(k+1)n}).
\end{aligned}
\tag{57}
$$

The second term on the right-hand-side of (57) can be bounded as

$$
\|\frac{1}{\theta\alpha}(\bar{z}^{kn} - \bar{z}^{(k+1)n})\| = \frac{1}{n^2\theta^2\alpha^2} \|\sum_{j=1}^{n} z_j^{kn} - z_j^{(k+1)n}\|^2 \leq \frac{1}{n\theta^2\alpha^2} \|z^{kn} - z^{(k+1)n}\|^2.
\tag{58}
$$

To bound the first term, it is noted that $2 \leq j \leq n$,

$$
z_{\tau_k(\ell)}^{kn+j-1} = \begin{cases} z_{\tau_k(\ell)}^{kn} + \frac{1}{\theta}(z_{\tau_k(\ell)}^{(k+1)n} - z_{\tau_k(\ell)}^{kn}), & 1 \leq \ell \leq j-1; \\ z_{\tau_k(\ell)}^{kn}, & \ell > j-1. \end{cases}
\tag{59}
$$

By (29), we have

$$
\begin{aligned}
&\|(I - \alpha \nabla f_{\tau_k(j)})(x^{kn+j-1}) - (I - \alpha \nabla f_{\tau_k(j)})(x^{kn})\|^2 \\
&= \|(I - \alpha \nabla f_{\tau_k(j)}) \circ \mathbf{prox}_{\alpha r}(\bar{z}^{kn+j-1}) - (I - \alpha \nabla f_{\tau_k(j)}) \circ \mathbf{prox}_{\alpha r}(\bar{z}^{kn})\|^2 \\
&\leq \|\bar{z}^{kn+j-1} - \bar{z}^{kn}\|^2 = \|\frac{1}{n} \sum_{\ell=1}^{n} (z_\ell^{kn+j-1} - z_\ell^{kn})\|^2 \\
&= \frac{1}{n^2\theta^2} \|\sum_{\ell=1}^{j-1} (z_\ell^{(k+1)n} - z_\ell^{kn})\|^2 \leq \frac{j-1}{n^2\theta^2} \sum_{\ell=1}^{j-1} \|z_\ell^{(k+1)n} - z_\ell^{kn}\|^2 \\
&\leq \frac{j-1}{n^2\theta^2} \|z^{(k+1)n} - z^{kn}\|^2.
\end{aligned}
\tag{60}
$$

Therefore we have

$$
\|\frac{1}{n\alpha}\sum_{j=1}^{n}\left((I-\alpha\nabla f_{\tau_k(j)})(x^{kn+j-1})-(I-\alpha\nabla f_{\tau_k(j)})(x^{kn})\right)\|^2
$$

$$
=\frac{1}{n^2\alpha^2}\|\sum_{j=2}^{n}\left((I-\alpha\nabla f_{\tau_k(j)})(x^{kn+j-1})-(I-\alpha\nabla f_{\tau_k(j)})(x^{kn})\right)\|^2
$$

$$
=\frac{1}{n^2\alpha^2}\sum_{j=2}^{n}\sqrt{j-1}\sum_{j=2}^{n}\frac{1}{\sqrt{j-1}}\|\left((I-\alpha\nabla f_{\tau_k(j)})(x^{kn+j-1})-(I-\alpha\nabla f_{\tau_k(j)})(x^{kn})\right)\|^2
$$

$$
\leq\frac{1}{n^2\alpha^2}\sum_{j=2}^{n}\sqrt{j-1}\sum_{j=2}^{n}\frac{1}{\sqrt{j-1}}\frac{j-1}{n^2}\|z^{(k+1)n}-z^{kn}\|^2
$$

$$
\leq\frac{4}{9}\frac{1}{n\theta^2\alpha^2}\|z^{(k+1)n}-z^{kn}\|^2. \tag{61}
$$

In the last inequality, we use the algebraic inequality that $\sum_{j=2}^{n}\sqrt{j-1}\leq\int_{1}^{n}\sqrt{x}dx=\frac{2}{3}x^{\frac{3}{2}}|_{1}^{n}\leq\frac{2}{3}n^{\frac{3}{2}}$.

Combining (58) and (61), we immediately obtain

$$
\min_{g\in\partial r(x^{kn})}\|\nabla F(x^{kn})+g\|^2\leq\|\nabla F(x^{kn})+\tilde{\nabla}r(x^{kn})\|^2
$$

$$
\leq(\frac{2}{3}+1)\Big(\frac{3}{2}\|\frac{1}{n\alpha}\sum_{j=1}^{n}\left((I-\alpha\nabla f_{\tau_k(j)})(x^{kn+j-1})-(I-\alpha\nabla f_{\tau_k(j)})(x^{kn})\right)\|^2
$$

$$
+\|\frac{1}{\theta\alpha}(\bar{z}^{kn}-\bar{z}^{(k+1)n})\|^2\Big)
$$

$$
\leq\frac{5}{3}(\frac{2}{3}\frac{1}{n\theta^2\alpha^2}\|z^{kn}-z^{(k+1)n}\|^2+\frac{1}{n\theta^2\alpha^2}\|z^{kn}-z^{(k+1)n}\|^2)
$$

$$
=\left(\frac{5}{3\alpha L}\right)^2\frac{L^2}{n\theta^2}\|z^{kn}-z^{(k+1)n}\|^2
$$

$$
\leq\left(\frac{5}{3\alpha L}\right)^2\frac{L^2}{\theta(1-\theta)(k+1)}\frac{1}{n}\|z^0-z^\star\|^2. \tag{62}
$$

$\square$

# E    Proof of Theorem 3

*Proof.* Before proving Theorem 3, we establish the epoch operator $\mathcal{S}_\pi$ and $\mathcal{S}_\tau$ are contractive in the following sense:

**Lemma 6.** *Under Assumption 2, if step size* $0<\alpha\leq\frac{2}{\mu+L}$*, it holds that*

$$
\|\mathcal{S}_\pi u-\mathcal{S}_\pi v\|_\pi^2\leq\left(1-\frac{2\theta\alpha\mu L}{\mu+L}\right)\|u-v\|_\pi^2 \tag{63}
$$

$$
\mathbb{E}\|\mathcal{S}_\tau u-\mathcal{S}_\tau v\|^2\leq\left(1-\frac{2\theta\alpha\mu L}{\mu+L}\right)\|u-v\|^2 \tag{64}
$$

$\forall u, v\in\mathbb{R}^{nd}$*, where* $\theta\in(0,1)$ *is the damping parameter.*

*Proof of Lemma 6.* For $\pi$-order cyclic sampling, without loss of generality, it suffices to show the case of $\pi = (1, 2, \ldots, n)$. To begin with, we first check the operator $\mathcal{T}_i$. Suppose $\boldsymbol{u}, \boldsymbol{v} \in \mathbb{R}^{nd}$,

$$
\|\mathcal{T}_i \boldsymbol{u} - \mathcal{T}_i \boldsymbol{v}\|_{h_i}^2
$$

$$
= \frac{1}{n} \sum_{j \neq i} \left( \mathrm{mod}_n (j - i - 1) + 1 \right) \|u_j - v_j\|^2
$$

$$
+ \|(I - \alpha \nabla f_i) \circ \mathbf{prox}_{\alpha r}(\mathcal{A}\boldsymbol{u}) - (I - \alpha \nabla f_i) \circ \mathbf{prox}_{\alpha r}(\mathcal{A}\boldsymbol{v})\|^2
$$

$$
\overset{(f)}{\leq} \frac{1}{n} \sum_{j \neq i} \left( \mathrm{mod}_n (j - i - 1) + 1 \right) \|u_j - v_j\|^2 + \left( 1 - \frac{2\alpha\mu L}{\mu + L} \right) \|\mathcal{A}\boldsymbol{u} - \mathcal{A}\boldsymbol{v}\|^2
$$

$$
\leq \frac{1}{n} \sum_{j \neq i} \left( \mathrm{mod}_n (j - i - 1) + 1 \right) \|u_j - v_j\|^2 + \frac{1}{n} \sum_{j=1}^n \|u_j - v_j\|^2 - \frac{2\alpha\mu L}{\mu + L} \|\mathcal{A}\boldsymbol{u} - \mathcal{A}\boldsymbol{v}\|^2
$$

$$
= \frac{1}{n} \sum_{j=1}^n \left( \mathrm{mod}_n (j - i) + 1 \right) \|u_j - v_j\|^2 - \frac{2\alpha\mu L}{\mu + L} \|\mathcal{A}\boldsymbol{u} - \mathcal{A}\boldsymbol{v}\|^2
$$

$$
= \|\boldsymbol{u} - \boldsymbol{v}\|_{h_{i-1}}^2 - \frac{2\alpha\mu L}{\mu + L} \|\mathcal{A}\boldsymbol{u} - \mathcal{A}\boldsymbol{v}\|^2. \tag{65}
$$

where $h_i$-norm in the first equality is defined as (28) with $\| \cdot \|_{h_0}^2 = \| \cdot \|_{h_0}^2 = \| \cdot \|_\pi^2$ and inequality (f) holds because

$$
\|x - \alpha \nabla f_i(x) - y + \alpha \nabla f_i(y)\|^2
$$

$$
= \|x - y\|^2 - 2\alpha \langle x - y, \nabla f_i(x) - \nabla f_i(y) \rangle + \alpha^2 \|\nabla f_i(x) - \nabla f_i(y)\|^2
$$

$$
\leq \left( 1 - \frac{2\alpha\mu L}{\mu + L} \right) \|x - y\|^2 - \left( \frac{2\alpha}{\mu + L} - \alpha^2 \right) \|\nabla f_i(x) - \nabla f_i(y)\|^2
$$

$$
\leq \left( 1 - \frac{2\alpha\mu L}{\mu + L} \right) \|x - y\|^2, \quad \forall x \in \mathbb{R}^d, y \in \mathbb{R}^d \tag{66}
$$

where the last inequality holds when $\alpha \leq \frac{2}{\mu + L}$. Furthermore, the inequality (66) also implies that

$$
\|[\mathcal{T}_i \boldsymbol{u}]_i - [\mathcal{T}_i \boldsymbol{v}]_i\|^2 = \|(I - \alpha \nabla f_i) \circ \mathbf{prox}_{\alpha r}(\mathcal{A}\boldsymbol{u}) - (I - \alpha \nabla f_i) \circ \mathbf{prox}_{\alpha r}(\mathcal{A}\boldsymbol{v})\|^2
$$

$$
\leq \left( 1 - \frac{2\alpha\mu L}{\mu + L} \right) \|\mathcal{A}\boldsymbol{u} - \mathcal{A}\boldsymbol{v}\|^2. \tag{67}
$$

where $[\,\cdot\,]_i$ denotes the $i$-th block coordinate.

Combining (65) and (67), we reach

$$
\|\mathcal{T}_i \boldsymbol{u} - \mathcal{T}_i \boldsymbol{v}\|_{h_i}^2 \leq \|\boldsymbol{u} - \boldsymbol{v}\|_{h_{i-1}}^2 - \frac{\eta(\alpha)}{1 - \eta(\alpha)} \|[\mathcal{T}_i \boldsymbol{u}]_i - [\mathcal{T}_i \boldsymbol{v}]_i\|^2 \tag{68}
$$

where $\eta(\alpha) = \frac{2\alpha\mu L}{\mu + L}$. With (68) and the following fact

$$
\|[\mathcal{T}_\pi \boldsymbol{u}]_j - [\mathcal{T}_\pi \boldsymbol{v}]_j\|^2 = \|[\mathcal{T}_j \cdots \mathcal{T}_2 \mathcal{T}_1 \boldsymbol{u}]_j - [\mathcal{T}_j \cdots \mathcal{T}_2 \mathcal{T}_1 \boldsymbol{v}]_j\|^2, \tag{69}
$$

we have

$$
\|\mathcal{T}_\pi \boldsymbol{u} - \mathcal{T}_\pi \boldsymbol{v}\|_\pi^2 \leq \|\mathcal{T}_{n-1} \cdots \mathcal{T}_1 \boldsymbol{u} - \mathcal{T}_{n-1} \cdots \mathcal{T}_1 \boldsymbol{v}\|_{h_{n-1}}^2 - \frac{\eta(\alpha)}{1 - \eta(\alpha)} \|[\mathcal{T}_\pi \boldsymbol{u}]_n - [\mathcal{T}_\pi \boldsymbol{v}]_n\|^2
$$

$$
\leq \|\mathcal{T}_{n-2} \cdots \mathcal{T}_1 \boldsymbol{u} - \mathcal{T}_{n-2} \cdots \mathcal{T}_1 \boldsymbol{v}\|_{h_{n-2}}^2 - \frac{\eta(\alpha)}{1 - \eta(\alpha)} \sum_{i=n-1}^n \|[\mathcal{T}_\pi \boldsymbol{u}]_i - [\mathcal{T}_\pi \boldsymbol{v}]_i\|^2
$$

$$
\leq \cdots
$$

$$
\leq \|\boldsymbol{u} - \boldsymbol{v}\|_{h_0}^2 - \frac{\eta(\alpha)}{1 - \eta(\alpha)} \sum_{i=1}^n \|[\mathcal{T}_\pi \boldsymbol{u}]_i - [\mathcal{T}_\pi \boldsymbol{v}]_i\|^2
$$

$$
= \|\boldsymbol{u} - \boldsymbol{v}\|_\pi^2 - \frac{\eta(\alpha)}{1 - \eta(\alpha)} \|\mathcal{T}_\pi \boldsymbol{u} - \mathcal{T}_\pi \boldsymbol{v}\|^2
$$

$$
\leq \|\boldsymbol{u} - \boldsymbol{v}\|_\pi^2 - \frac{\eta(\alpha)}{1 - \eta(\alpha)} \|\mathcal{T}_\pi \boldsymbol{u} - \mathcal{T}_\pi \boldsymbol{v}\|_\pi^2 \tag{70}
$$

where the last inequality holds because $\|\boldsymbol{u} - \boldsymbol{v}\|^2 \geq \|\boldsymbol{u} - \boldsymbol{v}\|_\pi^2, \forall \boldsymbol{u}, \boldsymbol{v} \in \mathbb{R}^{nd}$.

With (70), we finally reach

$$\|\mathcal{T}_\pi \boldsymbol{u} - \mathcal{T}_\pi \boldsymbol{v}\|_\pi^2 \leq \left(1 - \frac{2\alpha\mu L}{\mu + L}\right) \|\boldsymbol{u} - \boldsymbol{v}\|_\pi^2. \tag{71}$$

In other words, the operator is a contraction with respect to the $\pi$-norm. Recall that $\mathcal{S}_\pi = (1 - \theta)I + \theta\mathcal{T}_\pi$, we have

$$\begin{aligned}
\|\mathcal{S}_\pi \boldsymbol{u} - \mathcal{S}_\pi \boldsymbol{v}\|_\pi^2 &\leq (1 - \theta)\|\boldsymbol{u} - \boldsymbol{v}\|_\pi^2 + \theta\|\mathcal{T}_\pi \boldsymbol{u} - \mathcal{T}_\pi \boldsymbol{v}\|_\pi^2 \\
&\leq (1 - \theta)\|\boldsymbol{u} - \boldsymbol{v}\|_\pi^2 + \theta\left(1 - \frac{2\alpha\mu L}{\mu + L}\right)\|\boldsymbol{u} - \boldsymbol{v}\|_\pi^2 \\
&= \left(1 - \frac{2\theta\alpha\mu L}{\mu + L}\right)\|\boldsymbol{u} - \boldsymbol{v}\|_\pi^2. \tag{72}
\end{aligned}$$

As to random reshuffling, we use a similar arguments while replacing $\|\cdot\|_\pi^2$ by $\|\cdot\|^2$. With similar arguments to (68), we reach that

$$\|\mathcal{T}_i \boldsymbol{u} - \mathcal{T}_i \boldsymbol{v}\|_h^2 \leq \|\boldsymbol{u} - \boldsymbol{v}\|_{M_i h}^2 - \frac{\eta(\alpha)}{1 - \eta(\alpha)} h(i)\|[\mathcal{T}_i \boldsymbol{u}]_i - [\mathcal{T}_i \boldsymbol{v}]_i\|^2 \tag{73}$$

for any $h \in \mathbb{R}^d$ with positive elements, where $h$-norm follows (47) and $M_i$ follows (49). Furthermore, it follows direct induction that $\left(M_{\tau(i+1)} \cdots M_{\tau(n)} \mathbb{1}_n\right)(\tau(i)) = (1 + \frac{1}{n})^{n-i-1} \geq 1$, and we have

$$\begin{aligned}
&\|\mathcal{T}_{\tau(i)} \cdots \mathcal{T}_{\tau(1)} \boldsymbol{u} - \mathcal{T}_{\tau(i)} \cdots \mathcal{T}_{\tau(1)} \boldsymbol{v}\|_{M_{\tau(i+1)} \cdots M_{\tau(n)} \mathbb{1}_n} \\
\leq &\|\mathcal{T}_{\tau(i-1)} \cdots \mathcal{T}_{\tau(1)} \boldsymbol{u} - \mathcal{T}_{\tau(i-1)} \cdots \mathcal{T}_{\tau(1)} \boldsymbol{v}\|_{M_{\tau(i)} M_{\tau(i+1)} \cdots M_{\tau(n)} \mathbb{1}_n} \\
&- \frac{\eta(\alpha)}{1 - \eta(\alpha)} \left(M_{\tau(i+1)} \cdots M_{\tau(n)} \mathbb{1}_n\right)(\tau(i))\|[\mathcal{T}_{\tau(i)} \cdots \mathcal{T}_{\tau(1)} \boldsymbol{u}]_{\tau(i)} - [\mathcal{T}_{\tau(i)} \cdots \mathcal{T}_{\tau(1)} \boldsymbol{v}]_{\tau(i)}\|^2 \\
\leq &\|\mathcal{T}_{\tau(i-1)} \cdots \mathcal{T}_{\tau(1)} \boldsymbol{u} - \mathcal{T}_{\tau(i-1)} \cdots \mathcal{T}_{\tau(1)} \boldsymbol{v}\|_{M_{\tau(i)} M_{\tau(i+1)} \cdots M_{\tau(n)} \mathbb{1}_n} \\
&- \frac{\eta(\alpha)}{1 - \eta(\alpha)}\|[\mathcal{T}_{\tau(i)} \cdots \mathcal{T}_{\tau(1)} \boldsymbol{u}]_{\tau(i)} - [\mathcal{T}_{\tau(i)} \cdots \mathcal{T}_{\tau(1)} \boldsymbol{v}]_{\tau(i)}\|^2 \tag{74}
\end{aligned}$$

Therefore, with the fact that

$$\|[\mathcal{T}_\tau \boldsymbol{u}]_{\tau(i)} - [\mathcal{T}_\tau \boldsymbol{v}]_{\tau(i)}\|^2 = \|[\mathcal{T}_{\tau(i)} \cdots \mathcal{T}_{\tau(2)} \mathcal{T}_{\tau(1)} \boldsymbol{u}]_{\tau(i)} - [\mathcal{T}_{\tau(i)} \cdots \mathcal{T}_{\tau(2)} \mathcal{T}_{\tau(1)} \boldsymbol{v}]_{\tau(i)}\|^2 \tag{75}$$

it holds that

$$\begin{aligned}
&\|\mathcal{T}_\tau \boldsymbol{u} - \mathcal{T}_\tau \boldsymbol{v}\|^2 \\
= &\|\mathcal{T}_{\tau(n)} \cdots \mathcal{T}_{\tau(1)} \boldsymbol{u} - \mathcal{T}_{\tau(n)} \cdots \mathcal{T}_{\tau(1)} \boldsymbol{v}\|_{\mathbb{1}_n}^2 \\
\leq &\|\mathcal{T}_{\tau(n-1)} \cdots \mathcal{T}_{\tau(1)} \boldsymbol{u} - \mathcal{T}_{\tau(n-1)} \cdots \mathcal{T}_{\tau(1)} \boldsymbol{v}\|_{M_{\tau(n)} \mathbb{1}_n}^2 - \frac{\eta(\alpha)}{1 - \eta(\alpha)}\|[\mathcal{T}_\tau \boldsymbol{u}]_n - [\mathcal{T}_\tau \boldsymbol{v}]_{\tau(n)}\|^2 \\
\leq &\|\mathcal{T}_{\tau(n-2)} \cdots \mathcal{T}_{\tau(1)} \boldsymbol{u} - \mathcal{T}_{\tau(n-2)} \cdots \mathcal{T}_{\tau(1)} \boldsymbol{v}\|_{M_{\tau(n-1)} M_{\tau(n)} \mathbb{1}_n}^2 \\
&- \frac{\eta(\alpha)}{1 - \eta(\alpha)} \sum_{i=n-1}^n \|[\mathcal{T}_\tau \boldsymbol{u}]_{\tau(i)} - [\mathcal{T}_\tau \boldsymbol{v}]_{\tau(i)}\|^2 \tag{76} \\
\leq &\cdots \\
\leq &\|\boldsymbol{u} - \boldsymbol{v}\|_{M_{\tau(1)} \cdots M_{\tau(n)} \mathbb{1}_n}^2 - \frac{\eta(\alpha)}{1 - \eta(\alpha)} \sum_{i=1}^n \|[\mathcal{T}_\tau \boldsymbol{u}]_{\tau(i)} - [\mathcal{T}_\tau \boldsymbol{v}]_{\tau(i)}\|^2 \\
= &\|\boldsymbol{u} - \boldsymbol{v}\|_{M_{\tau(1)} \cdots M_{\tau(n)} \mathbb{1}_n}^2 - \frac{\eta(\alpha)}{1 - \eta(\alpha)}\|\mathcal{T}_\tau \boldsymbol{u} - \mathcal{T}_\tau \boldsymbol{v}\|^2.
\end{aligned}$$

Taking expectation on both sides and use the fact (52), we reach

$$\mathbb{E}_\tau \|\mathcal{T}_\tau \boldsymbol{u} - \mathcal{T}_\tau \boldsymbol{v}\|^2 \leq \|\boldsymbol{u} - \boldsymbol{v}\|^2 - \frac{\eta(\alpha)}{1 - \eta(\alpha)} \mathbb{E}_\tau \|\mathcal{T}_\tau \boldsymbol{u} - \mathcal{T}_\tau \boldsymbol{v}\|^2. \tag{77}$$

The left part to show contraction of $\mathcal{S}_\tau$ in expectation is the same as (72). $\qquad\square$

Based on Lemma 6, we are able to prove Theorem 3. When samples are drawn via $\pi$-order cyclic sampling, recall that $\boldsymbol{z}^{kn} = \mathcal{S}_\pi \boldsymbol{z}^{(k-1)n}$ and $\boldsymbol{z}^\star = \mathcal{S}_\pi \boldsymbol{z}^\star$, we have

$$\|\boldsymbol{z}^{kn} - \boldsymbol{z}^\star\|_\pi^2 \leq \left(1 - \frac{2\theta\alpha\mu L}{\mu + L}\right)^k \|\boldsymbol{z}^0 - \boldsymbol{z}^\star\|_\pi^2. \tag{78}$$

The corresponding inequality for random reshuffling is

$$\mathbb{E}\|\boldsymbol{z}^{kn} - \boldsymbol{z}^\star\|^2 \leq \left(1 - \frac{2\theta\alpha\mu L}{\mu + L}\right)^k \|\boldsymbol{z}^0 - \boldsymbol{z}^\star\|^2. \tag{79}$$

Notice that

$$
\begin{aligned}
\|x^{kn} - x^\star\|^2 =& \|\mathbf{prox}_{\alpha r}(\mathcal{A}\boldsymbol{z}^{kn}) - \mathbf{prox}_{\alpha r}(\mathcal{A}\boldsymbol{z}^\star)\|^2 \\
\leq& \|\mathcal{A}\boldsymbol{z}^{kn} - \mathcal{A}\boldsymbol{z}^\star\|^2 \\
\leq& \begin{cases} \frac{\log(n)+1}{n}\|\boldsymbol{z}^{kn} - \boldsymbol{z}^\star\|_\pi^2 & \text{for } \pi\text{-order cyclic sampling} \\ \frac{1}{n}\|\boldsymbol{z}^{kn} - \boldsymbol{z}^\star\|^2 & \text{for random reshuffling.} \end{cases}
\end{aligned} \tag{80}
$$

Combining (80) with (78) and (79), we reach

$$(\mathbb{E})\,\|x^{kn} - x^\star\|^2 \leq \left(1 - \frac{2\theta\alpha\mu L}{\mu + L}\right)^k C \tag{81}$$

where

$$C = \begin{cases} \frac{\log(n)+1}{n}\|\boldsymbol{z}^{kn} - \boldsymbol{z}^\star\|_\pi^2 & \text{for } \pi\text{-order cyclic sampling} \\ \frac{1}{n}\|\boldsymbol{z}^{kn} - \boldsymbol{z}^\star\|^2 & \text{for random reshuffling.} \end{cases}$$

$\square$

**Remark 5.** *One can taking expectation over cyclic order $\pi$ in 16 to obtain the convergence rate of Prox-DFinito shuffled once before training begins*

$$\mathbb{E}\|x^{kn} - x^\star\|^2 \leq \left(1 - \frac{2\theta\alpha\mu L}{\mu + L}\right)^k C$$

*where $C = \frac{(n+1)(\log(n)+1)}{2n^2}\|\boldsymbol{z}^0 - \boldsymbol{z}^\star\|^2$.*

# F   Optimal Cyclic Order

*Proof.* We sort all $\{\|z_i^0 - z_i^\star\|^2\}_{i=1}^n$ and denote the index of the $\ell$-th largest term $\|z_i^0 - z_i^\star\|^2$ as $i_\ell$. The optimal cyclic order $\pi^\star$ can be represented by $\pi^\star = (i_1, i_2, \cdots, i_{n-1}, i_n)$. Indeed, due to sorting inequality, it holds for any arbitrary fixed order $\pi$ that $\|\boldsymbol{z}^0 - \boldsymbol{z}^\star\|_{\pi^\star}^2 = \sum_{\ell=1}^n \frac{\ell}{n}\|z_{i_\ell} - z_{i_\ell}^\star\|^2 \leq \sum_{\ell=1}^n \frac{\ell}{n}\|z_{\pi(\ell)} - z_{\pi(\ell)}^\star\|^2 = \|\boldsymbol{z}^0 - \boldsymbol{z}^\star\|_\pi^2$. $\square$

# G   Adaptive importance reshuffling

**Proposition 6.** *Suppose $\boldsymbol{z}^{kn}$ converges to $\boldsymbol{z}^\star$ and $\{\|z_i^0 - z_i^\star\|^2 : 1 \leq i \leq n\}$ are distinct, then there exists $k_0$ such that*

$$\min_{g \in \partial\, r(x^{kn})} \|\nabla F(x^{kn}) + g\|^2 \leq \frac{CL^2}{(k+1-k_0)\theta(1-\theta)}$$

*where $\theta \in (0, 1)$ and $C = \left(\frac{2}{\alpha L}\right)^2 \frac{\log(n)+1}{n}\|\boldsymbol{z}^{k_0} - \boldsymbol{z}^\star\|_{\pi^\star}^2$.*

*Proof.* Since $\boldsymbol{z}^{kn}$ converges to $\boldsymbol{z}^\star$, we have $w^{k+1}$ converges to $\|z^0 - \boldsymbol{z}^\star\|^2$. Let $\epsilon = \frac{1}{2}\min\{|\|z_i^0 - z_i^\star\|^2 - \|z_j^0 - z_j^\star\|^2| : 1 \leq i \neq j \leq n\}$, then there exists $k_0$ such that $\forall\, k \geq k_0$, it holds that $|w^k(i) - \|z_i^0 - z_i^\star\|^2| < \epsilon$ and hence the order of $\{w^k(i) : 1 \leq i \leq n\}$ are the same as $\pi^\star$ for all $k \geq k_0$. Further more by the same argument as Appendix D.2, we reach the conclusion. $\square$

Table 2: Step sizes suggested by best known analysis

| Step size | DFinito | SVRG | SAGA |
|---|---|---|---|
| RR | $\frac{2}{L+\mu}$ | $\frac{1}{\sqrt{2}Ln}$ if $n \geq \frac{2L}{\mu} \frac{1}{1-\frac{\mu}{\sqrt{2}L}}$ else $\frac{1}{2\sqrt{2}Ln}\sqrt{\frac{\mu}{L}}$ [18] | $\frac{\mu}{11L^2n}$ [37] |
| Cyc. sampling | $\frac{2}{L+\mu}$ | $\frac{1}{4Ln}\sqrt{\frac{\mu}{L}}$ [18] | $\frac{\mu}{65L^2\sqrt{n(n+1)}}$ [25] |

# H  Best known guaranteed step sizes of variance reduction methods under without-replacement sampling

# I  Existence of highly heterogeneous instance

**Proposition 7.** *Given sample size $n$, strong convexity $\mu$, smoothness parameter $L$ $(L > \mu)$, step size $\alpha$ and initialization $\{z_i^0\}_{i=1}^n$, there exist $\{f_i\}_{i=1}^n$ such that $f_i(x)$ is $\mu$-strongly convex and $L$-smooth with fixed-point $z^\star$ satisfying $\|z^0 - z^\star\|_{\pi^\star}^2 = \mathcal{O}(\frac{1}{n})\|z^0 - z^\star\|^2$.*

*Proof.* Let $f_i(x) = \frac{1}{2}\|A_i x - b_i\|^2$, we show that one can obtain a desired instance by letting $\lambda = \mu$ and choosing proper $A_i \in \mathbb{R}^{k \times d}$, $b_i \in \mathbb{R}^k$.

First we generate a positive number $\beta = q^2 \in (0,1)$ and vectors $\{t_i \in \mathbb{R}^d : \|t_i\| = \sqrt{n}\}_{i=1}^n$. Let $v = \frac{1}{n}\sum_{i=1}^n (z_i^0 - q^{i-1}t_i)$. Then we generate $A_i \in \mathbb{R}^{k \times d}$ such that $\mu I \preceq A_i^T A_i \preceq LI$, $1 \leq i \leq n$, which assures $f_i$ are $\mu$-strongly convex and $L$-smooth.

After that we solve $A_1^T \delta_1 = \frac{1}{\alpha}(z_i^0 - v - q^{i-1}t_i)$, $\forall\, 1 \leq i \leq n$. Note these $\{\delta_i\}_{i=1}^n$ exist as long as we choose $k \geq d$. Therefore we have $\sum_{i=1}^n A_i^T \delta_i = \sum_{i=1}^n \frac{1}{\alpha}(z_i^0 - v - q^{i-1}t_i) = 0$ by our definition of $v$.

Since $\nabla f_i(x) = A_i^T(A_i x - b_i)$, then by letting $b_i = A_i v + \delta_i$, it holds that

$$
\begin{aligned}
\nabla F(x) &= \frac{1}{n}\sum_{i=1}^n \nabla f_i(x) \\
&= \frac{1}{n}\sum_{i=1}^n A_i^T(A_i x - b_i) \\
&= \frac{1}{n}\sum_{i=1}^n A_i^T A_i(x - v) - \frac{1}{n}\sum_{i=1}^n A_i^T \delta_i \\
&= \frac{1}{n}\sum_{i=1}^n A_i^T A_i(x - v)
\end{aligned}
\tag{82}
$$

and hence we know $\nabla F(v) = 0$, i.e., $x^\star = v$ is a global minimizer.

We finally follow Remark 1 to reach

$$
\begin{aligned}
z_i^\star &= (I - \alpha\nabla f_i)(x^\star) \\
&= v - \alpha A_i^T(A_i v - b_i) \\
&= v + \alpha A_i^T \delta_i \\
&= z_i^0 - q^{i-1}t_i
\end{aligned}
\tag{83}
$$

and hence $\|z_i^0 - z_i^\star\|^2 = q^{2(i-1)}\|t_i\|^2 = n\beta^{i-1}$. By direct computation, we can verify that $\|z^0 - z^\star\|_{\pi^\star}^2 = \mathcal{O}(\frac{1}{n})\|z^0 - z^\star\|^2$. $\qquad\square$

# J More experiments

**DFinito vs SGD vs GD.** In this experiment, we compare DFinito with full-bath gradient descent (GD) and SGD under RR and cyclic sampling (which is analyzed in [22]). We consider the regularized logistic regression task with MNIST dataset (see Sec. 5.1). In addition, we manipulate the regularized term $\frac{\lambda}{2}\|x\|^2$ to achieve cost functions with different condition numbers. Each algorithm under RR is averaged across 8 independent runs. We choose optimal constant step sizes for each algorithm using the grid search. In Fig. 3, it is observed that SGD under without-replacement sampling works, but

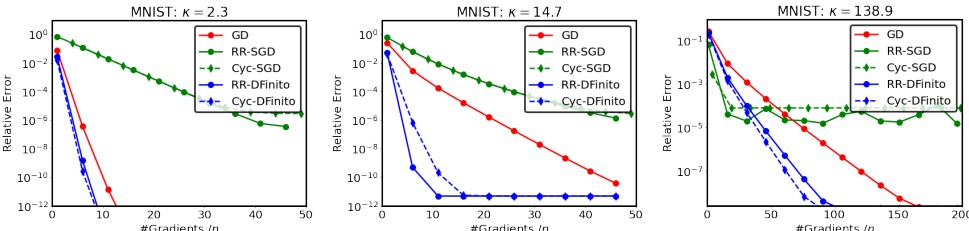

Figure 3: Comparison between DFinito, SGD and GD over MNIST across different conditioning scenarios. The relative error indicates $(\mathbb{E})\|\nabla x - x^\star\|^2/\|x^0 - x^\star\|^2$.

it will get trapped to a neighborhood around the solution and cannot converge exactly. In contrast, DFinito will converge to the optimum. It is also observed that DFinito can outperform GD in all three scenarios, and the superiority can be significant for certain condition numbers. While this paper establishes that the DFinito under without-replacement sampling shares the same theoretical gradient complexity as GD (see Table 1), the empirical results in Fig. 3 imply that DFinito under without-replacement sampling may be endowed with a better (though unknown) theoretical gradient complexity than GD. We will leave it as a future work.

**Influence of various data heterogeneity on optimal cyclic sampling.** According to Proposition 5, the performance of DFinito with optimal cyclic sampling is highly influenced by the data heterogeneity. In a highly data-heterogeneous scenario, the ratio $\rho = \mathcal{O}(1/n)$. In a data-homogeneous scenario, however, it holds that $\rho = \mathcal{O}(1)$. In this experiment, we examine how DFinito with optimal cyclic sampling converges with varying data heterogeneity. To this end, we construct an example in which the data heterogeneity can be manipulated artificially and quantitatively. Consider a problem in which each $f_i(z) = \frac{1}{2}(a_i^T x)^2$ and $r(x) = 0$. Moreover, we generate $A = \text{col}\{a_i^T\} \in \mathbb{R}^{n \times d}$ according to the uniform Gaussian distribution. We choose $n = 100$ and $d = 200$ in the experiment, generate $p_0 \in \mathbb{R}^d$ with each element following distribution $\mathcal{N}(0, \sqrt{n})$, and initialize $z_i^0 = p_0/\sqrt{c}$, $1 \le i \le c$ otherwise $z_i^0 = 0$. Since $x^\star = 0$ and $\nabla f_i(x^\star) = 0$, we have $z_i^\star = 0$. It then holds that $\|\mathbf{z}^0 - \mathbf{z}^\star\|^2 = \sum_{i=1}^c \|z_i^0\|^2 = \|p_0\|^2$ is unchanged across different $c \in [n]$. On the other hand, since $\|\mathbf{z}^0 - \mathbf{z}^\star\|_{\pi^\star}^2 = \sum_{i=1}^c \frac{i}{n} \frac{\|p_0\|^2}{c} = \mathcal{O}(\frac{c}{n}\|p_0\|^2)$, we have $\rho = \mathcal{O}(c/n)$. Apparently, ratio $\rho$ ranges from $\mathcal{O}(1/n)$ to $\mathcal{O}(1)$ as $c$ increases from 1 to $n$. For this reason, we can manipulate the data heterogeneity by simply adjusting $c$. We also depict DFinito with random-reshuffling (8 runs' average) as a baseline. Figure 4 illustrates that the superiority of optimal cyclic sampling vanishes gradually as the data heterogeneity decreases.

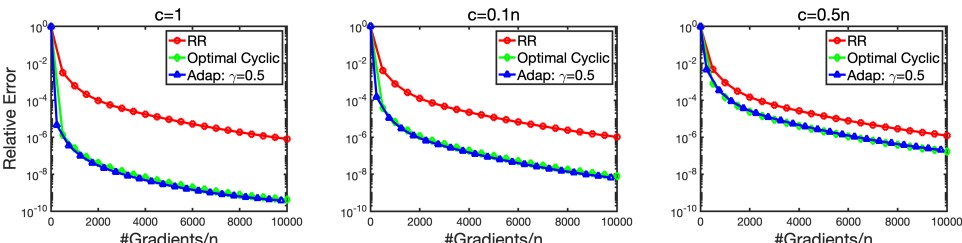

Figure 4: Comparison between various sampling fashions of DFinito across different data-heterogeneous scenarios. The relative error indicates $(\mathbb{E})\|\nabla F(x)\|^2/\|\nabla F(x^0)\|^2$.

**Comparison with empirically optimal step sizes.** Complementary to Sec. 5.1, we also run experiments to compare variance reduction methods under uniform sampling (US) and random reshuffling (RR) with empirically optimal step sizes by grid search, in which full gradient is computed once per two epochs for SVRG. We run experiments for regularized logistic regression task with CIFAR-10 ($\kappa = 405$), MNIST ($\kappa = 14.7$) and COVTYPE ($\kappa = 5.5$), where $\kappa = L/\mu$ is the condition number. All algorithms are averaged through 8 independent runs. From Figure 5, it is observed that all three

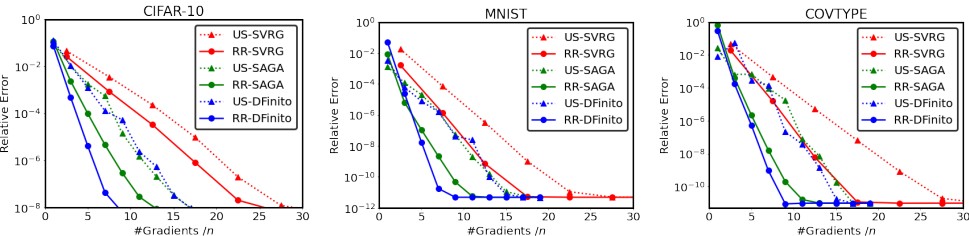

Figure 5: Comparison of variance reduced methods under with/without-replacement sampling. The relative error indicates $(\mathbb{E})\|x - x^\star\|^2/\|x^0 - x^\star\|^2$.

variance reduced algorithms achieve better performance under RR, and DFinito outperforms SAGA and SVRG under both random reshuffling and uniform sampling. While this paper and all other existing results listed in Table 1 (including [18]) establish that the variance reduced methods with random reshuffling have worse theoretical gradient complexities than uniform-iid-sampling, the empirical results in Fig. 5 imply that variance reduced methods with random reshuffling may be endowed with a better (though unknown) theoretical gradient complexity than uniform-sampling. We will leave it as a future work.