# OpenReview forum: "An Improved Analysis and Rates for Variance Reduction under Without-replacement Sampling Orders "
_NeurIPS.cc/2021/Conference — NeurIPS 2021 Poster_

### Official Review · Reviewer_zgXx · 2021-07-06

**Rating:** 7
**Confidence:** 4

**Summary:**

In this paper, the authors study a stochastic variance reduced method under sampling rules that are without replacement for smooth and non-smooth, convex and strongly convex objectives. More precisely :
- they develop a proximal method call Prox-DFinito for which they study convergence rates for random reshuffling, cyclic and shuffling once samplings
- in the cyclic sampling, they derive an optimal fixed ordering (and a practical adaptive variant which do not require the knowledge of $z^*$
- Finally, the authors highlight the fact that, when assuming data heterogeneity, their study lead to a convergence rate independent of the number of data samples. This was up until now just known for iid sampling.



**Limitations And Societal Impact:**

- Authors do not clearly explain how to set $\theta$ (except in the strongly convex case) and how this affects the performance. As said before, the reader don't get why and if this parameter is important or if it is just introduce for the analysis.
- Same remark for $\gamma$ in (22)
- Numerical experiments seem to correctly recover what theory predicts for data-heterogeneity case and for recovering GD rate


**COMMENT AFTER THE REBUTTAL**

I thank the authors for there clear and exhaustive answers to my questions/concerns and those of the other reviewers.

I read the authors' rebuttal and it made things clearer in my mind on some points :
- the extended Table 1 clarifying the comparison with other works, especially [18]
- I read carefully the discussion about the measure of suboptimality done in this work (pointed out by reviewer y5Mf). I also think it is a limitation of this paper.
- After reading answers to my review and the one of Reviewer VeN3, I now understand now why the dampening parameter is required.

I this paper will be valuable to the community both for its theoretical results (sample independent rate with highly heterogeneous assumption) and its technical aspects (eg the introduction of order-specific norms). Also, fair comparison with previous work (especially [18] in/out of the Big data regime) will be clearly presented in the final version of this work. **Thus, I increase my score from 6 to 7.**



Side comment : I would have been interested to see what is the magnitude of $\rho$ for different datasets and if it is close to $\mathcal{O} (1/n)$ for  some real data sets.

**Main Review:**

First of all, the paper is very well written (especially the background detailing precious works), very very few typos, main proofs seem correct.

Minor comments which are just my opinion:
- 1) I do not find that "generally convex" is a proper way to describe a function which is just "convex". It's not standard and reading it quickly might make the reader think it's more general than convexity.
- 2) Lines 37 and 39 : i would not start a sentence with a ref, I would write "Authors of [XX] establish ..."
- 3) Prop. 1 & 3: I would not put "(proof in Appendix XX)" inside the proposition but before or after.


Minor/medium importance comments:
- 4) Could you precise in Table 1 that you assume that the $f_i$'s are strongly convex (not $F$), it makes easier the comparison with [18]
- 5) Eq. (13) : I don't know what is the squared norm of a subdifferential. I think it's not standard. I would rewrite this to make it more mathematically correct if you meant, "the squared norm of all elements in the subdifferential".
- 6) Remark 3: typo "One can takING" -> "take"
- 7) maybe it is worth citing again [16] in Remark 3.
- 8) line 154: why "Apparently" ? It makes the sentence look potentially false
- 9) line 232, 2 typos: $==$ and a word is missing before the parenthesis
- 10) line 242, typo : written $\nabla f (x^*)$, it should be $\nabla f_i (x^*)$ instead

Major comments:
- 11) Could you clearly explain and highlight the fact that Table 1 is comparing number of gradients evaluations (total complexity) to reach $\epsilon$-solutions, it makes easier the comparison with Table 1 in[18]
- 12) I find that the rate of [18] are missing in table 1. Even if an entire section in the appendix describe the differences, we need a more detailed comparison in the main text instead of the short 3 lines at the end of subsection 1.2. Giving the pros and cons of each method (and clearly stating the difference in memory $\mathcal{O} (nd)$ (yours) vs $\mathcal{O} (d)$ ([18])).
- 13) Lines 98-100: It's not clearly explained how you came up with this damping parameter. Why do you need it ?

Major comments on Appendix K:
- 14) You do not discuss the "Big Data regime" + strong convexity for which the authors of [18] clearly have a gain : in their terms they get an iteration complexity of $\mathcal{O} (\frac{L}{\mu} \log (\frac{1}{\epsilon}))$. This assumption lead to the same total complexity than yours for RR and worst cyclic order : $\mathcal{O} (n \frac{L}{\mu} \log (\frac{1}{\epsilon}))$ right ?
- 15) Could you precise that your memory cost is $n$ times the one for SVRG ?

Questions :
- 16) Did you come with the idea of this "order-specific norm". If not, could you please add a reference ?
- 17) I was not aware of the "data heterogeneity assumption" before reading this paper. Where does it come from ? Could you add ref about this ?  Is it met in practice ? What is the link between the assumption and the heterogeneity of the data ?
- 18) Proof of theorem 1, line 465: Why is $z_{\pi(j)}^{kn+j-1} = z_{\pi(j)}^{kn}, \ \forall j=1, \ldots, n\$ true ? For instance, if $\pi(j)=j$ and $k=0$, then it would mean that
$$z_j^{j-1} = z_j^0$$
which is not necessarily the case if the index sampled at step $j-1$ is j. Maybe I'm wrong on this one, but let me know if there is an easy way to see this.

**Time Spent Reviewing:**

9

---

> ### Author Response · Authors · 2021-08-10
> **Response to Reviewer zgXx**
>
> We thank the reviewer for the positive comments. We have attempted to address all the questions as best as we can. We will be glad to clarify any further comments.
>
> **Minor comments:** Many thanks for the careful reviews!
>
> 1. We will change the "Generally-convex" terminology to "Convex" terminology.
>
> 2. Thanks for the writing suggestion! We will revise it accordingly.
>
> 3. We will "(proof in Appendix XX)" before each lemma, theorem, or proposition.
>
> **Medium comments:** Many thanks for the careful reviews!
>
> 4. We have created a new comparison table based on all reviewers' comments. Please check Table 1 in rebuttal response Part II to Reviewer y5Mf, in which we have explicitly added one column to indicate whether each algorithm is analyzed under the individual strong convexity, or the averaged strong convexity.
>
> 5. Yes, we meant the squared norm of all elements in the subdifferential set. In fact, we found the equivalent definition $\min\limits_{g \in \partial r(x)}\|\nabla F(x)+g\|^2$ mentioned by the reviewer y5Mf has much less ambiguity. We will use it in our revised draft.
>
> 6. Revised. Thanks for the correction!
>
> 7. Thanks for the great suggestion. We will cite [16] in Remark 3 and clarifies the relations to it.
>
> 8. We will add the following clarifications: Given $n$ samples, there will be $n!$ without-replacement orders to draw samples. Let $\tau_k$ denote the sampling order used in the $k$-th epoch. If we assume $\tau_k$ will take each sampling order with equal probability, then it is a uniformly distributed random variable with $n!$ realizations.
>
> 9. Revised. Thanks for the careful review!
>
> 10. Agree. Revised.
>
> **Major comments:**
>
> 11. Done. We have made it clear in the caption of the newly-created Table 1 listed in the rebuttal response Part II to Reviewer y5Mf.
>
> 12. Done. We have put the rates in [18] into the newly-created Table 1 listed in in rebuttal response Part II to Reviewer y5Mf. We also explicitly listed the memory cost.
>
> 13. Thanks for this valuable question.
>
>     - As we discussed with Reviewer VeN3 in the first question, the construction of the damping step is motivated by the technical challenges we met when establishing the convergence rate of Prox-DFinito. Our proof idea comes from the fixed-point recursion and operator theory, see. e.g., [Bauschke and Combettes, 2011], which guarantees that an averaged non-expansive operator will generate sequences that has $O(1/k)$ sub-linear convergence rate.
>
>     - In our proof, we first introduce an operator $\mathcal{T}$ (see Sec.2.1) and then formulate Prox-Dfinito method with without-replacement sampling orders into a fixed-point iteration $\mathcal{z}^{(k+1)n} = \mathcal{T}\mathcal{z}^{(kn)}$. We were able to prove that the operator $\mathcal{T}$ is non-expansive (see Lemma 1), but we were stuck to further prove it is an **averaged** non-expansive operator. But if we introduce a new operator $\mathcal{S} = (1-\theta) I + \theta \mathcal{T}$ (which is essentially the well-known Krasnoselskii-Mann Iteration, also known as damping), it is shown in [Bauschke and Combettes, 2011] that the operator $\mathcal{S}$ will be an averaged non-expansive operator. This is the fundamental reason why we introduce the damping step in the algorithm design. We hope this discussion can provide new insights to the reviewer.
>
>     [Bauschke and Combettes, 2011] Bauschke, Heinz H., and Patrick L. Combettes. Convex analysis and monotone operator theory in Hilbert spaces. Vol. 408. New York: Springer, 2011.
>
> **Major comments on Appendix K:**
>
> 14. Yes. For the big data regime, the strongly-convex results in [18] is improved to the same complexity $O(n\frac{L}{\mu}\log(\frac{1}{\epsilon}))$ as random-reshuffling results. We have added it to the newly-created Table 1 in the rebuttal response Part II to Reviewer y5Mf.
>
> 15. The memory cost for Prox-DFinito is $O(nd)$ while SVRG in [18] is $O(d)$. We clarified it in the newly-created Table 1 in the rebuttal response Part II to Reviewer y5Mf.
>
> **Questions:**
>
> 16. Yes. To our best knowledge, this paper provides the **first** result on an order-specific metric that can evaluate how cyclic sampling performs with different orders. Before us, literature can just evaluate the performance of the worst-case cyclic orders. We believe it is a useful contribution to the community on stochastic methods with without-replacement sampling.
>
> 17. Thanks for this valuable question. Here are our immature thougths:
>
>     - To better illustrate the data-heterogeneity metric introduced in (21), we simplified the setting by initializing $z_i^0$ close to $x^\star$ so that $\|z_i^0 - z_i^\star\|^2$ is proportional to $\|\nabla f_i(x^\star)\|^2$ (see Remark 5). Note that $\|\nabla f_i(x^\star)\|^2 = \|\nabla f_i(x^\star) - \nabla F(x^\star)\|^2$ (because $\nabla F(x^\star) = 0$ for smooth scenario) measures the data heterogeneity of function $f_i(x)$.
>
>     - In literatures such as [Koloskova et.al., 2020, Assumption 3a], authors use the metric $\frac{1}{n}\sum_{i=1}^n\|\nabla f_i(x^\star)\|^2$ to evaluate the averaged data heterogeneity of all individual functions.
>
>     - However, the metric introduced in (21), i.e., $\rho = \|\nabla \mathbf{f}(x^\star)\|^2_{\pi^\star}/\|\nabla \mathbf{f}(x^\star)\|^2_2 \in (\frac{1}{n}, 1)$ where $\nabla \mathbf{f}(x^\star) = [\nabla f_1(x^\star);\cdots;\nabla f_n(x^\star)]$, is more likely to gauge the balance between each $\|\nabla f_i(x^\star)\|^2$.  For example, in the balanced scenario where $\|\nabla f_i(x^\star)\|^2 \approx \|\nabla f_j(x^\star)\|^2$ for any $i$ and $j$, we have $\rho \approx \frac{n(n+1)}{2n^2} = O(1)$. However, in the highly unbalanced scenario where $\|\nabla f_i(x^\star)\|^2 = n\beta^{i-1}$ for some $\beta \in (0, 1)$ (see the discussion in Sec. 4.2 in the paper), we have $\rho = O(1/n)$. Condition (21) implies that the optimal cyclic sampling is more useful for the highly unbalanced data-heterogeneity scenario, i.e., $\rho = O(1/n)$.
>
>     - The metric $\rho$ introduced in (21) is novel, and it does not appear in previous literature. In real dataset such as Cifar-10 and MNIST, the value of $\rho$ typically lies between $O(\frac{1}{n})$ and $O(1)$, which indicates the optimal cyclic order is still useful, but its performance is not as significant as the what we show in simulation Fig. 2, see the experiments we present in the response to Reviewer dHKd.
>
> [Koloskova et.al., 2020] Koloskova et.al., A Unified Theory of Decentralized SGD with Changing Topology and Local Updates, ICML 2020.
>
> 18. The relation should be correct. Note that $\pi$ denotes a cyclic order that maps $\{1,2,\dots,n\}$ to $\{1,2,\dots,n\}$ in a one-to-one manner. It indicates that $\pi$ enforces the “sample-once-per-epoch” property. The sample $\pi(j)$ will only be sampled at the $j$-th iteration within epoch $kn$, and hence $z_{\pi(j)}$ is not updated for all iterations $kn, kn+1, \cdots, kn+j-1$. As a result, it holds that $z^{kn+j-1}_{\pi(j)} = z^{kn}_\pi(j)$. As to the example provided by the reviewer, if $\pi(j)=j$ and $k=0$, then the sample $j$ is only updated in the $j$-th update (not in the $j-1$-th update), therefore, $z^{0}_j=z^{1}_j=\cdots=z^{j-1}_j$ remain the same.
>
> **Limitations And Societal Impact:**
>
> 19. We hope our response in Q13 can help the reviewer understand why $\theta$ is needed in analysis. Besides, $\theta=\frac{1}{2}$ will minimize the upper bounds in (13-15).
>
> 20. It is difficult to justify the effect of the choice of $\gamma$ from our theoretical analysis. However from an empirical point of view, we find $\gamma=0.5$ typically achieves the most robust and fast convergence.
>
> 21. We thank the reviewer for the positive comments on our numerical experiments.
>
> Lastly, we will be glad to clarify any further comments.

---

> > ### Comment · Reviewer_zgXx · 2021-08-24
> > **Comment about the heterogenity assumption**
> >
> > In my **Comment after rebuttal**, I explained why I increased my score. I did read your detailed answers to all reviewers (especially your discussion with Reviewer y5Mf which where very enlightening).
> >
> > Now I just remember that I was wondering if you had a name in mind for condition (21) (which we understand on next page holds in the data heterogeneous case).

---

> > > ### Author Response · Authors · 2021-08-28
> > > **Thank you very much for increasing the rating! Much appreciated!**
> > >
> > > We thank the reviewer for increasing the rating. We have read the reviewer's final comments carefully.
> > >
> > > We are happy that our response has addressed most concerns of the reviewer. As to the limitation in the measure of the suboptimality, we will clearly acknowledge it in the revision. Below is our response to the new comments from the reviewer.
> > >
> > > - **Magnitude of $\rho$ in real datasets.** We have tested various real datasets such as MNIST and CIFAR10, and the value of $\rho$ is not very close to $O(1/n)$, but somewhere between $O(1/n)$ and $O(1)$. This indicates the optimal cyclic order is still useful for these real datasets, but its superiority is not as significant as in the constructed dataset we show in simulation Fig. 2. Please check the additional experiments we present in the response to Reviewer dHKd. We conjecture that $\rho = O(1/n)$ will hold more easily for distributed stochastic optimization (especially in the Federated Learning setting) where each different node will collect data that may follow different distributions. We are testing more real datasets, especially those used in Federated learning literatures, and will add them in the revised draft.
> > >
> > > - **Name of condition (21).** Condition (21) is novel and does not appear in previous literatures. Since it is more likely to gauge the balance between each $|\nabla f_i(x^\star)|^2$, we may name it as "Unbalanced data heterogeneity condition" to make it different from other assumptions used in, e.g.,  [Koloskova et.al., 2020]. We would like to hear feedback from the reviewer on such name and make it more precise in the revision.
> > >
> > > We thank the reviewer for all of these valuable discussions, which clearly improves the quality of our draft. We will carefully follow the reviewer's comments and suggestions when revising the paper.

---

### Official Review · Reviewer_VeN3 · 2021-07-13

**Rating:** 6
**Confidence:** 4

**Summary:**

This paper proposes a new first-order algorithm using without-replacement strategy to solve a class of composite convex minimization problems. The main idea is to modify the well-known scheme, called Finito/MISO method by applying a without-replacement strategy and a damping step. Under the convexity and L-smoothness of f, the proposed algorithm achieves O(1/k) rate in epoch on the optimality residual. When f is additionally strongly convex, the rate is improved to linear as in standard proximal gradient methods. The analysis relies on a weight norm defined through the order of the underlying shuffling strategy. The authors also compare their method with other schemes such as a coordinate descent method with cyclic rule, and standard GD. Next, the authors also investigate the optimal cyclic rule for sampling and propose an adaptive variant. Numerical examples on standard logistic regression problem are presented to illustrate the performance of the proposed methods.


**Ethics Review Area:**

["I don’t know"]

**Limitations And Societal Impact:**

The paper does not discuss the limitation as well as social impact.

**Main Review:**

Originality: I believe that the result of this paper is new, especially the use of a new weighted norm in this case. In fact, the algorithm can be cast into a variant of the ARock method [Peng et al, 2015], but the use of without-replacement is new. The use of damped step is not new, which has been widely used in fixed-point methods as well as in optimization. The key idea of analysis is to show that the corresponding fixed-point mapping is nonexpansive, which allows to use a damped step and achieve O(1/k) rate in epoch. However, the convergence rate is not really encouraging even it is comparable with GD. The reason is that this method requires to store n auxiliary vectors as in SAGA, making it less practical when n and d are large.

Quality/clarity: The paper is well written and well-motivated in general. It does have both algorithmic and theoretical contributions. The technicality sounds and the analysis seems to be nontrivial.

Significance: Overall, the paper has nice contribution in terms of new algorithm, especially when using without-replacement strategy.

Below are some concrete comments and questions:

-- Is the damped step in Algorithm 1 really the key to achieve the O(1/k) rate? In fact, it is not needed for the strongly convex case. I have a feeling that this is only needed if the fixed-point mapping is only non-expansive. Could the authors clarify and discuss this point, because it is highlighted in the paper?

-- It seems that the nonexpansiveness of T is rather trivial since it is constituted by a proximal gradient operator which is nonexpansive for such a chosen stepsize. Does it need any other properties to guarantee the non-expansiveness of T in (11)?

-- The comparison with SVRG and SAGA using randomized reshuffling rule seems to not make sense. These variants may not work with without-replacement rules unless an appropriate variant (e.g., in [18,37]) is used. The authors may use [18] for SVRG, but for SAGA, which variant is used in the experiments?

-- The weighted norm only depends on z0 and z*, and does not depend on the landscape of the problem. This is a very interesting aspect of the new method. Can you also add more discussion on this aspect to elaborate the new contribution?


**Time Spent Reviewing:**

10

---

> ### Author Response · Authors · 2021-08-10
> **Response to Reviewer VeN3**
>
> We thank the reviewer for the positive comments. We will cite ARock [Peng et.al., 2015] and clarify its relationship with our work. We have attempted to address all the questions as best as we can. We will be glad to clarify any further comments.
>
> 1. We thank the reviewer for bringing this interesting question.
>
>     - We share the same feeling with the reviewer, i.e., the damping step might not be needed in the convex scenario, as is the case in the strongly-convex scenario. However, we meet significant technical difficulties when removing the damping step. **Without the damping, we can prove $\mathcal{T}$ is a non-expansive operator, but not necessarily an $\alpha$-averaged non-expansive operator**. To construct an $\alpha$-averaged non-expansive operator, we introduce the damping step and let $\mathcal{S} = (1-\theta) I + \theta \mathcal{T}$. All the sublinear convergence rate is achieved with the help of operator $\mathcal{S}$.
>
>     - In the strongly convex scenario, we can directly prove that $\mathcal{T}$ is a contraction operator, which is the reason why we do not need damping for strongly-convex scenario.
>
>     - If we can prove $\mathcal{T}$ is an $\alpha$-averaged operator by itself for the convex scenario, then we do not need the damping step. However, we cannot prove this result at this moment.
>
>
> 2. We thank the reviewer for the interest in the proof details.
>
>     - The establishment of the non-expansiveness of $\mathcal{T}$ is non-trivial. Since Prox-DFinito is with the without-replacement sampling order, the operator $\mathcal{T}$ is a composition of $n$ consecutive element-wise-updated operators $\mathcal{T}_i$, see Proposition 2. **However, each $\mathcal{T}_i$ may not be non-expansive in the $\ell_2$-norm**. To see it, recall the definition of $\mathcal{T}_i$ in Sec. 2.1. While the backward operator $\mathrm{prox}(\cdot)$ and forward $I - \alpha \nabla f_i(\cdot)$ is non-expansive, the operator $z_i^{t+1} \leftarrow \mathcal{A} \mathcal{z}^t = \mathrm{average}(z_1^t, \cdots, z_n^t)$ will not be non-expansive if $z_i^t$ is less than the average $\bar{z}^t$ which will be assigned to $z_i^{t+1}$.
>
>     - While $\mathcal{T}$ is not non-expansive in the $\ell_2$-norm, we do construct an order-specific norm, i.e., the $\pi$-norm, with which $\mathcal{T}$ can be established to be non-expansive. The construction of such norm, and the proof are highly non-trivial, see Lemma 1 and its proof in Appendix C.1.
>
>     - In summary, the difficulty to establish the non-expansiveness of $\mathcal{T}$ lies in two aspects: (1) $\mathcal{T}$ is not non-expansive in $\ell_2$-norm due to the without-replacement sampling order, and (2) identification of the norm with which $\mathcal{T}$ can be shown non-expansive is highly non-trivial.
>
>
> 3. We used the variants of SVRG in [18] in our experiments. For SAGA with random reshuffling, we used the variant in [37]. For SAGA with cyclic sampling, we used the variant in [25]. As stated in Table 2 in the paper, all variants have convergence guarantee with corresponding step sizes.
>
>
> 4. Our immature thoughts on this phenomenon are as follows:
>
>     - As we discussed above, we first establish that $\mathcal{T}$ is a non-expansive operator (Lemma 1), and then we are able to prove the sub-linear (Lemma 2) or linear convergence rate of sequence $z^{kn}$. Finally, we utilize the convergence of $z^{kn}$ to bound the sequence of $x^{kn}$ (Theorems 1-3). It explains why we have $\|z^{0} - z^\star\|^2$ in the upper bounds.
>
>     - While $\|z^{0} - z^\star\|^2$ does not relate to the landscape $F(x)$, it does relate to the shape of gradient $\nabla F(x)$. To see it,  recall that $z_i^0 = x^0 - \alpha \nabla f_i(x^0)$ and $z_i^\star = x^\star - \alpha \nabla f_i(x^\star)$,  it holds that $\|z^{0} - z^\star\|^2 = \sum_{i=1}^n \|x^0 - \alpha \nabla f_i(x^0) - (x^\star - \alpha \nabla f_i(x^\star))\|^2 $. We are not sure whether this new discussion can bring any inspiration to the reviewer. We are more than happy for further discussions on this point.

---

> > ### Comment · Reviewer_VeN3 · 2021-08-22
> > **Thank you very much for your response. I decided to maintain my score**
> >
> > Dear authors:
> >
> > Thank you very much for your response and clarification.
> > After reading other reviewers' comments and the response, I find that though paper contains new results, there are some fundamental weaknesses such as the optimality condition used in the convex case, and the convergence rate in the non-strongly convex case. Note that the convergence rate is achieved in epoch-wise, where after each epoch every function component is used one. Therefore, the nonexpansiveness of $\mathcal{T}$ is not really surprised (of course it needs some technical details in the proofs). This usually leads to $\mathcal{O}(1/k)$ rate in the square norm of subgradient/gradient.  In my opinion, the most novelty of the paper is the new norm defined in Definition 1. To achieve convergence rate in the objective residual, the author may need to change the proof technique from fixed-point-based theory to convex optimization techniques. Therefore, I think my current score is reasonable and fair for this paper. I really appreciate the authors' effort in replying reviewers' comments.

---

### Official Review · Reviewer_y5Mf · 2021-07-16

**Rating:** 6
**Confidence:** 5

**Summary:**

The paper proposes a new method called Prox-DFinito based on the proximal Finito with without-replacement sampling. The authors derive complexity bounds for the proposed method in convex (for making the squared norm of the gradient small) and strongly convex cases (for making the squared distance to the solution small) that match under some additional assumptions the rate of Gradient Descent. Moreover, under additional assumptions on the objective function, the authors show sample-size independent bound in the convex case. The proofs are non-standard but clean and easy to follow. However, the paper has several strong weaknesses.

**Limitations And Societal Impact:**

The authors adequately addressed the limitations and potential negative societal impact of their work.

**Main Review:**

## Strengths

1. **Clarity.** The paper is clearly written and well-organized.

2. **Interesting proofs.** The proofs of the theoretical results of the paper are non-standard and, therefore, valuable. In addition, the proofs are easy to follow and do not contain inaccuracies.

## Weaknesses

1. **Weak guarantees in the convex case.** In the convex case (i.e., all $f_i$ are convex), the authors establish complexity bounds ensuring that $\text{dist}(\|\nabla F(x^{kn}) - \partial r(x^{kn})\|^2, 0) = \min_{g\in \partial r(x^{kn})}\|\nabla f(x^{kn}) + g\|^2 \leq \varepsilon$. In other papers and, in particular, in [18], bounds are obtained to ensure $F(x^{kn}) - F(x^*) \leq \varepsilon$ (no regularization). When the objective is convex and smooth it is possible that the gradient is small but functional suboptimality is huge. Moreover, it is more important to achieve small functional suboptimality rather than the small norm of the gradient. Although the proofs are interesting, the result in the convex case is not strong enough and cannot be directly compared with known results. The authors should clarify it in Table 1 since in the cited papers the bounds in the generally convex case are established for functional suboptimality, not for the squared gradient norm. Moreover, the authors should clarify what do they mean by such a comparison. In the non-proximal scenario, one can use classical inequality $\|F(x)\|^2 \le 2L(F(x) - F(x^{*}))$, but there are no analogs of such inequality for the composite problems. This place should be carefully clarified.

2. **Complexity bounds depend on $\frac{1}{n}\|\mathcal{z}^0 - \mathcal{z}^{*}\|_{\pi\text{ or }2}^2$.** This norm has implicit dependence on the heterogeneity of local loss functions $f_i$. Indeed, if $\mathcal{z}^0$ is close to $x^*$, then for RR $C \sim \frac{\alpha^2}{n}\sum_{i=1}^n\|\nabla f_i(x^{*})\|$ where $C$ is the factor appearing in all upper bounds.In other words, even if the starting point is extremely close to the solution the constant $C$ can be large. This observation immediately implies that all derived rates in the paper can be arbitrarily worse than known results. In contrast, the concurrent work [18] does not have this issue.

3. **Results are weaker than state-of-the-art ones.** In view of the first weakness, the complexity bounds in the convex case are weaker by default since they do not provide guarantees for the functional suboptimality. Next, in the strongly convex case, much tighter results are given in [35], where the authors use $L$ being the average of individual smoothness constants (that can be almost $n$ times smaller than the worst one) and $\mu$ being the strong convexity constant of the average and do not require individual loss functions to be strongly convex. In contrast, this paper uses the worst $L$ for all summands and relies on the assumption that all $f_i$ are $\mu$-strongly convex. Moreover, in view of weakness 2, it is even impossible to fairly compare the results without additional assumptions.

4. **Incomplete comparison with the related work.** The paper tries to create an impression that the obtained results are the current state-of-the-art. However, it is not true. As it was mentioned earlier, the derived results are weaker than the state-of-the-art ones but the authors do not write about such important details. Moreover, since [18] is the recent concurrent work, the authors should add more details in the main part of the paper. First of all, the rates from [18] should be added in Table 1 (otherwise it creates a biased impression). Next, the authors should explicitly write that the results in the convex case from [18] are stronger (see weakness 1). Moreover, [18] contains some results for the "Big Data regime" establishing $O(n \frac{L}{\mu} \log \frac{1}{\varepsilon})$ complexity in the strongly convex case and it is shown without assuming strong convexity of each summand. Finally, rates from [18] do not have the issue described in weakness 2. Therefore, the authors should write about all of these details in the paper and at least briefly mention them somewhere in the main part.

## Questions and comments

1. **Table 1, results from [33] and [5]:** The bounds are strange since their "physical dimension" is incorrect. That is, the complexity should be dimensionless quantity whereas $\frac{L^2}{\mu}$ and $\frac{L}{\mu^2}$ are not. Please, correct these bounds.

2. **line 42, "accelerated":** This word has a certain meaning (Nesterov's acceleration) in the optimization literature that differs from what the authors want to say (e.g., see d'Aspremont, Alexandre, Damien Scieur, and Adrien Taylor. "Acceleration methods." arXiv preprint arXiv:2101.09545 (2021).). The authors should rewrite the sentence.

3. **lines 46-47, "In the generally convex scenario, existing rates for without-replacement sampling with variance reduction are still far worse than GD":** this is not true because [18] exists.

4. **line 101, "Note that the damping step does not incur additional memory requirements":** In theory, it is still $O(nd)$, but in practice, there is a big difference between $nd$ and $2nd$.

5. **Algorithm 3:** The algorithm is not equivalent to Algorithm 1.

6. **inequality (54), the last line:** the identity is not correct since $\sum_{t=0}^{j-i-1} \neq \left(1 + \frac{1}{n}\right)^{j-i-1}$. However, one can stop one line earlier since it is not important for the next arguments in the proof.

## Comment after the rebuttal

I thank the authors for their detailed response. I have read it carefully.

**Bounds in the convex case.** Both authors and me agree that bounds for the norm of the gradient are weaker than bounds for functional suboptimality, so my concern is still valid. However, I acknowledge that before this work and [18], which is written independently and has different analysis, are the first results for VR methods with without-replacement sampling that match the rate of GD in the convex case under the assumption that all $f_i$ and $F$ have the same smoothness constant. Moreover, in the "highly-heterogeneous scenario," the rate derived in this paper is independent of $n$ that I find quite interesting.

**On the $\frac{1}{n}\|z^0 - z^*\|^2$.** Indeed, if $z_i^0 = x^0 - \alpha \nabla f_i(x^0)$, then there is no issue. I suggest the authors write about this initialization in the main text. I think it is important for the comparison with other results.

**Revised Table 1 and clarifications about the comparison with other works.** Now, these parts are much more transparent for the readers. I believe these corrections are very important.

**On Algorithm 3.** Thank you for the clarifications. Now I see the equivalence. Initially, I was confused because of the difference in the formulas for $z_{i_t}^t$ in Algorithms 1 and 3. However, after your clarifications, I realized that the two methods are indeed equivalent. I suggest the authors add these clarifications to the paper.

Overall, although the paper has several limitations, it does contain some valuable contributions. In particular, the idea of using order-specific norm in the analysis is of separate interest. Taking all these aspects into account, I decided to increase my score from 4 to 6.

**Time Spent Reviewing:**

12 hours

---

> ### Author Response · Authors · 2021-08-10
> **Response to Reviewer y5Mf. Part II: Rebuttal Details**
>
> **A revised Table 1 and some clarification remarks:**
>
> Table 1: Number of individual gradient evaluations needed by each algorithm to reach an$\epsilon$-accurate solution. Notation $\tilde{O}(·)$ hides logarithmic factors. Error metrics $(E)\|\nabla F(x)\|^2$ and $(E)\|x-x^\star\|^2$ are used for convex and strongly convex problems, respectively
>
> |Algorithm | Support Prox  | Sampling | Memory | Assumption$^{\star}$|  Convex | Strongly Convex|
> |:----:|:----:|:----:|:----:|:----:|:----:|:----:|
> |Proximal GD | Yes | Full-batch | $O(d)$ | $F(x)$ | $O(n\frac{L^2}{\epsilon})$ |$O(n\frac{L}{\mu}\log(\frac{1}{\epsilon}))$|
> |SGD [22]|  No | Cyclic | $O(d)$ | $f_i(x)$ |  $O(n(\frac{L}{\epsilon})^{\frac{3}{2}})$  | $O(n(\frac{L}{\mu})^\frac{3}{2}\frac{1}{\sqrt{\epsilon}})$  |
> |SGD [22]|  No | RR | $O(d)$ | $f_i(x)$ |  $O(\sqrt{n}(\frac{L}{\epsilon})^{\frac{3}{2}})$  | $O(\sqrt{n}(\frac{L}{\mu})^\frac{3}{2}\frac{1}{\sqrt{\epsilon}})$  |
> |PSGD [21] | Yes | RR | $O(d)$| $f_i(x)$ |$\diagdown$ | $O(n(\frac{L}{\mu})^\frac{3}{2}\frac{1}{\sqrt{\epsilon}})$ |
> |PIAG [35] | Yes | Cyclic/RR | $O(nd)$ | $F(x)$ |$\diagdown$  |$\tilde{O}(n\frac{L}{\mu}\log(\frac{1}{\epsilon}))$|
> |AVRG [37]| No | RR | $O(d)$ | $f_i(x)$ | $\diagdown$  | $O(n(\frac{L}{\mu})^2\log(\frac{1}{\epsilon}))$|
> |SAGA [33] | Yes | Cyclic | $O(nd)$ | $f_i(x)$ |$\tilde{O}(n^3\frac{L^2}{\epsilon})$ | $O(n^3\frac{L^2}{\mu^2}\log(\frac{1}{\epsilon}))$|
> |SVRG [37] | Yes | Cyclic | $O(d)$ | $f_i(x)$ | $O(n^3\frac{L^2}{\epsilon})$ | $O(n^3\frac{L^2}{\mu}\log(\frac{1}{\epsilon}))$|
> | DIAG [23] | No | Cyclic | $O(nd)$ | $f_i(x)$ |$\diagdown$ | $O(n\frac{L}{\mu}\log(\frac{1}{\epsilon}))$|
> |Cyc. Cord. Update [5] | Yes | Cyclic/RR| $O(d)$ |$f_i(x)$ |$O(n2^n\frac{L^2}{\epsilon})$ | $O({n}\frac{L^3}{\mu^3}\log(\frac{1}{\epsilon}))$ |
> | SVRG [18]$^{\diamond}$ | No | Cyclic | $O(d)$ | $F(x)$ |$O(n\frac{L^2}{\epsilon})$ | $O(n(\frac{L}{\mu})^\frac{3}{2}\log(\frac{1}{\epsilon}))$|
> | SVRG [18]$^{\diamond}$ | No | RR | $O(d)$ | $f_i(x)$ |$O(n\frac{L^2}{\epsilon})$ | $O(\sqrt{n}(\frac{L}{\mu})^\frac{3}{2}\log(\frac{1}{\epsilon}))^{\dagger}$|
> | SVRG [18]$^{\diamond}$ | No | RR | $O(d)$ | $F(x)$ |$O(n\frac{L^2}{\epsilon})$ | $O(n\frac{L}{\mu}\log(\frac{1}{\epsilon}))^{\dagger}$|
> | SVRG [18]$^{\diamond}$ | No | RR | $O(d)$ | $F(x)$ |$O(n\frac{L^2}{\epsilon})$ | $O(n(\frac{L}{\mu})^\frac{3}{2}\log(\frac{1}{\epsilon}))$|
> | Prox-DFinito (ours) | Yes | RR | $O(nd)$ | $f_i(x)$ | $O(n\frac{L^2}{\epsilon})$ | $O(n\frac{L}{\mu}\log(\frac{1}{\epsilon}))$|
> | Prox-DFinito (ours) | Yes | worst cyclic | $O(nd)$ | $f_i(x)$ | $\tilde{O}(n\frac{L^2}{\epsilon})$ | $\tilde{O}(n\frac{L}{\mu}\log(\frac{1}{\epsilon}))$|
> | Prox-DFinito (ours) | Yes | best cyclic | $O(nd)$ | $f_i(x)$ | $\tilde{O}(\frac{L^2}{\epsilon})$ | $\tilde{O}(n\frac{L}{\mu}\log(\frac{1}{n\epsilon}))$|
>
> $\diamond$. [18] is an **independent** and **concurrent** work.
>
> $\star$. The "Assumption" column indicates the scope of smoothness and strong convexity, where $F(x)$ means smoothness and strong convexity in the average sense. In contrast, $f_i(x)$ means assuming smoothness and strong convexity for each summand function.
>
> $\dagger$. Big data regime: $n>O(\sqrt{\frac{L}{\mu}})$.
>
> **Remark 1**. Except for reference [5] and our Prox-DFinito algorithm whose convergence analyses are based on the **gradient norm**, results in other references are based on **function value** metric $F(x^{kn}) - F(x^\star)$  (i.e., objective error). The function value metric can imply the gradient norm metric, but not always vice versa. To compare [5] and Prox-DFinito in the same metric, we have to transform the results in other references into the gradient norm metric. The comparison is listed in Table 1. When the gradient norm metric is used, we observe that the rates of Prox-DFinito match that with gradient descent, and are state-of-the-art compared to the existing results. However, the rate of Prox-DFinito in terms of the function value is not known yet (this unknown rate may end up being worse than those of the other methods).
>
> **Remark 2**. Except for references [18, 35] and proximal-GD algorithm whose convergence analyses are conducted by assuming the average of each function to be $\bar{L}$-smooth (and perhaps $\bar{\mu}$-strongly convex), results in other references are based on a stronger assumption that each function to be $L$-smooth (and perhaps $\mu$-strongly convex). Note that $\bar{L}$ can be much smaller than $L$ in practice. To compare [18, 35] and proximal-GD with other references under the same assumption, we let each $L_i = L$ and hence $\bar{L} = L$ in Table 1. However, it is worth noting that when each $L_i$ is drastically different from each other and can be evaluated precisely, the results relying on $\bar{L}$ (e.g., [35] and [18]) can be much better than the other references.
>
>
> **Rebuttal details:**
>
> 1. Thanks for the comments!
>
>     - We agree that the gradient norm is a generally (but not always) weaker metric than the function value. We apologize for not clarifying that the other references are using the function value metric. Since the function value result of Prox-DFinito is unknown yet, we have to transform the results in other references into the gradient norm metric so that Prox-DFinito and [25] can be comparable to them. We have clarified that in the revised Table 1 and Remark 1, see above.
>
>     - For the non-smooth scenario, $\mathrm{dist}(\|\nabla F(x^{kn}) + \partial\,r(x^{kn})\|^2,0)$ may not be bounded by the functional suboptimality, and hence the our Prox-DFinito results are not comparable with those in [21, 25, 35, 37]. The results listed in Table 1 are all for the smooth scenario of [21, 25, 35, 37], and we use **Support Prox** to indicate whether the results cover the non-smooth scenario or not.
>
>
> 2. We believe the dependence on $(1/n)\|z^0 - z^\star\|^2_{\pi\ \mathrm{or}\ 2}$ is **not an issue** for Prox-DFinito.
>
>     - In Remark 5 in the paper, we initialize $z^0 = x^\star$ to provide an interpretation that $\|z^0 - z^\star\|^2$ can be regarded as an importance indicator. However, it does not mean we have to initialize $z^0 = x^\star$. In fact, with (4a) and (9) in the paper, we know $z_i^t = x^{t} - \alpha \nabla f_i(x^{t})$ and $z_i^\star = x^\star - \alpha \nabla f_i(x^\star)$. When $x^0$ is initialized as $x^\star$ as suggested by the reviewer, we should initialize $z_i^0 = x^\star - \alpha \nabla f_i(x^\star) = z_i^\star$, rather than $x^\star$. This implies that when $x^0$ is very close to $x^\star$, $z^0$ is also close to $z^\star$. In this sense, the dependence on $\|z^0 - z^\star\|^2$ should not be an issue.
>
>     - On the other hand, for any arbitrary $x^0$, if we initialize  $z_i^0 = x^0 - \alpha \nabla f_i(x^0)$, it holds that $\|z_i^0 - z_i^\star\|^2 \le \|x^0 - x^\star\|^2$ due to the non-expansiveness of $I - \alpha \nabla f_i(\cdot)$. This implies that $\frac{1}{n}\|z^0 - z^\star\|^2 \le \|x^0 - x^\star\|^2 = O(1)$. As a result, the initialization condition for Prox-DFinito and [18] should not make a big difference in convergence rate.
>
>
> 3. Many thanks for the comments! We agree that a unified $L$ assumption is stronger than that in [18] and [35]. We have clarified the assumption used by each method in the above Table 1, and we added a clarifying Remark 2 (see above) to make it clear that when each $L_i$ is drastically different and can be evaluated precisely, the results in [18] and [35] can be much better. However, the assumption on the unified L and $\mu$ is not uncommon in literature, see [21, 22, 23, 37, 33, 5].
>
>
> 4. Many thanks for the comments!
>
>     - We have added [18] into the above Table 1, and clarified that [18] utilizes function value metric and assume the average of each individual function to be $\bar{L}$-smooth, which are better than our results. We also clarified [18] has an improved result in the big-data regime. We will carefully discuss [18] in the main body of the revised draft.
>
>     - While [18] has impressive results, we have to emphasize that [18] is a work that appeared at the same time as our work, and the results are complementary to ours. Our paper provides new results on a non-smooth scenario, an order-specific norm to gauge the influence of the cyclic-sampling order, and a new sample-size-independent order in a certain scenario. Providing competitive sampling complexity is one of our contributions.
>
> **Other comments:** Thanks for the careful review. We appreciate these questions.
>
> 1. We have revised it in Table 1 above.
>
> 2. We will replace the word "accelerate" with "speed up"
>
> 3. It is true. We will explicitly acknowledge the contribution and impressive results provided by [18]. However, before [18] and our work, rates for without-replacement sampling with variance reduction are indeed far worse than GD.
>
> 4. If Algorithm 3 is correct, DFinito only requires $(n+1)d$ memories since $d_i$ is removed immediately after being used by local update. Otherwise, DFinito will need $2nd$ memories.
>
> 5. In Algorithm 3, each $z_i$ is damped right after the gradient $d_i$ is updated. In Algorithm 1, all $\{z_i\}_{i=1}^n$ are damped at the end of each epoch. We believe it is the only difference between Algorithm 1 and Algorithm 3,  and it does not cause essential changes as $z_i$ would not be used any more after being updated per epoch. Could the reviewer please show us the potential mistake so that we can better resolve your concerns?
>
> 6. Thanks for pointing this typo out. We do miss the factor $\binom{j-i-1}{t-2}$ caused by the all possible combinations of $\{j_2< j_3< \cdots< j_{t-1}\}$ in the third line of (54). We have corrected it.

---

> > ### Comment · Reviewer_y5Mf · 2021-08-15
> > **Thank you a lot for the detailed response. I am increasing my score to 6**
> >
> > I thank the authors for their detailed response. I have read it carefully.
> >
> > **Bounds in the convex case.** Both authors and me agree that bounds for the norm of the gradient are weaker than bounds for functional suboptimality, so my concern is still valid. However, I acknowledge that before this work and [18], which is written independently and has different analysis, are the first results for VR methods with without-replacement sampling that match the rate of GD in the convex case under the assumption that all $f_i$ and $F$ have the same smoothness constant. Moreover, in the "highly-heterogeneous scenario," the rate derived in this paper is independent of $n$ that I find quite interesting.
> >
> > **On the $\frac{1}{n}\|z^0 - z^*\|^2$.** Indeed, if $z_i^0 = x^0 - \alpha \nabla f_i(x^0)$, then there is no issue. I suggest the authors write about this initialization in the main text. I think it is important for the comparison with other results.
> >
> > **Revised Table 1 and clarifications about the comparison with other works.** Now, these parts are much more transparent for the readers. I believe these corrections are very important.
> >
> > **On Algorithm 3.** Thank you for the clarifications. Now I see the equivalence. Initially, I was confused because of the difference in the formulas for $z_{i_t}^t$ in Algorithms 1 and 3. However, after your clarifications, I realized that the two methods are indeed equivalent. I suggest the authors add these clarifications to the paper.
> >
> > Overall, although the paper has several limitations, it does contain some valuable contributions. In particular, the idea of using order-specific norm in the analysis is of separate interest. Taking all these aspects into account, I decided to increase my score from 4 to 6.

---

> > > ### Author Response · Authors · 2021-08-16
> > > **Many thanks for increasing the rating! Much appreciated!**
> > >
> > > We thank the reviewer very much for increasing the rating! We have read the reviewer's final comments carefully.
> > >
> > > 1. **Bounds in the convex case.** We really appreciate the reviewer can find our results valuable to the community in spite of its several limitations. We will explicitly highlight these limitations in the revised draft. We will also highlight the impressive results that [18] achieve on variance-reduction under without-replacement sampling.
> > >
> > >
> > > 2. **On the $\frac{1}{n}\|z^0-z^\star\|^2$.** We are glad our response has addressed the reviewer's concerns.
> > >
> > >
> > > 3. **Revised Table 1 and clarification about the comparison with other works.** We agree that Table 1 listed above is more transparent now. We will use this table in the revised draft.
> > >
> > >
> > > 4. **On Algorithm 3.** We are glad our our response has addressed the reviewer's confusions. We will add these clarifications in the revised draft.
> > >
> > > We thank the reviewer for all these valuable discussions, which clearly improves the quality of our draft.  We will carefully follow the reviewer's comments and suggestions when revising the paper.

---

> ### Author Response · Authors · 2021-08-10
> **Response to Reviewer y5Mf. Part I: Rebuttal Summary**
>
> Thanks for the valuable comments! Below we address all your questions. We will be glad to clarify any further comments.
>
> **Rebuttal Summary**:
>
> As the reviewer (and other reviewers) commented, our analysis technique is novel. It provides valuable new insights to the community that studies stochastic methods with without-replacement sampling orders. In addition to giving the nearly-SOTA$^*$ sample complexity compared to existing results, we believe our analysis has two unique contributions to the community:
>
> - We introduce an order-specific norm to gauge how a specific cyclic order affects the convergence rate and sample complexity. Previous and contemporaneous analyses evaluate the performance of the worst cyclic order, not a specific cyclic order.
>
> - Our result shows when Prox-DFinito using without-replacement sampling achieves a complexity independent of the sample size. This result sheds lights on the **long-standing** open question of whether the sample complexity bounds of without-replacement variance-reduction methods can match those of their uniform-sampling counterparts
>
> The limitations of our analysis, as the reviewer pointed out, are the use of gradient norm as the convergence metric and the assumption that each local function $f_i(x)$ has a unified Lipschitz constant $L$ (and $\mu$ when we also assume strong convexity). We have revised our paper to highlight these limitations. To address these issues,
>
> - We have created a new comparison table (see Part II below) and explicitly added the assumption and memory cost to each method;
>
> - We explicitly remarked that the comparison utilizes the gradient-norm metric, while other references use the function value as their metrics;
>
> - We add the results in the contemporaneous work [18] into the comparison table;
>
> - We will rephrase one of our contributions as "When the gradient norm metric is used and each local function $f_i(x)$ has a unified Lipschitz constant $L$ (and a unified constant $\mu$ if strongly convex), our rates match with gradient descent and are state-of-the-art compared to existing results for without-replacement sampling with variance-reduction. However, results in [18] (and others) are based on the function value metric, which is preferable, and [18, 35] assume different $L_i$ and $\mu_i$ in their setting. If these metrics or assumptions are utilized, our results may be worse than [18, 35]". We would like to hear feedback from the reviewer on such rephrase and make it more precise in the revision.
>
> We agree that there are functions where they can have large objective errors but tiny gradients (or other first-order residuals). For these functions, the gradient norm is a weaker metric than the objective error. That said, gradients (and sub/gradient-based first-order residuals) play important roles and are worth looking at. They are used to stop a method since objective errors are generally unavailable. In strongly convex (and certain non-strongly convex) models and in a few special but popular nonconvex models, gradient norms bound the distances to solutions and objective errors. Many first-order splitting methods (e.g., primal-dual hybrid gradient, ADMM) have monotone fixed-point residual but non-monotone objective values, so they are typically analyzed with gradients and subgradients; their objective rates are derived from the rates of fixed-point residuals. Therefore, while we agree objective rates are preferable, gradient-norm rates are still good convergence indicators.
>
> As to point 2 by the reviewer, we believe it is not an issue and provide a detailed response. Please check the detail in Part II.
>
> In summary, we have updated our paper to clarify limitations in the convergence metric and assumptions and added the comparison table. However, we hope the reviewer can understand that these metric and assumptions are not deal-breakers. Our new insights on specific cyclic orders are obtained out of the metric despite the limitations. Other existing analyses have not produced any "order-specific-norm" and "sample-size independent order" to our knowledge.
>
> The rebuttal details will be provided in Part II.
>
> $^*$ Nearly-SOTA: When we use the gradient norm metric and the unified L-smoothness assumption, our rates match with gradient descent and are state-of-the-art compared to existing results. However, if the function value metric is used, or the L-smoothness assumption is utilized, the rate of Prox-Finito is not known yet and can be worse than SOTA rates (such as [18] or [35]).

---

### Official Review · Reviewer_dHKd · 2021-07-20

**Rating:** 7
**Confidence:** 3

**Summary:**

This paper develops a proximal damped version of the Finito algorithm. The algorithm is proved to achieve the same convergence rate as proximal GD, for cyclic sampling, random reshuffling and shuffling-once versions of the algorithm. Further, the authors claim that this is the first* shuffling based variance reduction algorithm to achieve the convergence rate. The paper also gives a new norm that captures the optimality of sampling orders and provides a heuristic based on that for importance based reshuffling.

Besides the theoretical results, the empirical results seem to suggest that the proposed algorithm is indeed faster than other variance reduction algorithms.

*: The authors cite a concurrent work (Malinovsky et al.) that also achieves the same convergence rates for general convex functions, but the algorithms in the two papers are different.

**Limitations And Societal Impact:**

1. Please move the comparison done in Table 3, appendix K to the main paper.
2. Please also provide plots for the test error of the evaluations done on MNIST, CIFAR-10 etc.

**Main Review:**

This paper proposes Prox-DFinito, which is a shuffling based variance reduction algorithm.

The theoretical results show that the cyclic sampling, random reshuffling and shuffling-once versions of the algorithm achieve the same convergence rate as GD (up to logarithmic factors) on general convex and strongly convex functions. The paper also proposes a new heuristic to get good sampling orders based on a new norm.

Overall, the theoretical results look good and the empirical evaluation support the theory.

A concern that I have is regarding the claim that the optimal cyclic sampling can achieve a sample complexity of $\widetilde{O}(L^2/\epsilon)$, which is independent of $n$. While this might indeed be correct, this might be slightly misleading. The fact is that without looking at all the $n$ functions at least once, good convergence cannot be achieved - consider the case where a fraction of the functions have minima very far away from others. Hence, even to determine the optimal cyclic order, at least $\Omega(n)$ computation must be done. The authors should clarify this.

The empirical evaluation for the optimal cycling sampling order seems to be done on an artificial dataset of quadratics that share the same minima. This does not give sufficient indication to whether the proposed optimal cyclic sampling order would work in practice. Can the authors provide evaluation on real datasets?

**Time Spent Reviewing:**

5

---

> ### Author Response · Authors · 2021-08-10
> **Response to Reviewer dHKd**
>
> Many thanks for the positive comments! We have attempted to address all the questions as best as we can.
>
> **Major comments:**
>
> * [Sample-size independent order] We thank the reviewer for bring this question, which does need further clarifications.
>     - In Sec.4.1, we established a sample-size independent order $\tilde{O}(L^2/\epsilon)$ only **when the data-heterogeneity condition (21) holds and the optimal cyclic order is known in advance**.
>
>     - When $\epsilon$ is sufficiently small so that $L^2/\epsilon > n$, the complexity $\tilde{O}(L^2/\epsilon)$ implies that we have to take more then $n$ samples to achieve an $\epsilon$-accurate solution, and hence, we will definitely go through all samples at least once.
>
>     - When $\epsilon$ is sufficiently large so that $L^2/\epsilon < n$, Prox-DFinito does not need to check all samples given the optimal cyclic order $\pi^\star$ and under condition (21). Intuitively speaking, condition (21) tells us that the data is highly unbalanced heterogeneous, and some $f_i(x)$ is very important to optimize, while the others have been very close to the global minima. In addition, the optimal cyclic order specifics that the we will first sample important $f_i(x)$'s to optimize, and then sample less important ones. With condition (21) and the optimal cyclic order, the error will quickly drop below $L^2/\epsilon$ after all important $f_i(x)$'s are optimized, and there is no need to go through all $n$ functions. On the other hand, cyclic sampling with an arbitrary cyclic order, or random reshuffling, will have to take roughly $O(n)$ samples to reach the desired accuracy because it is difficult for them to find the important $f_i(x)$'s in advance.
>
>     - When the optimal cyclic order is not known, we do have to go through all functions and incur a computation  cost $\Omega(n)$.
>
>     We hope these explanations can clarify the reviewer's confusions, and we are happy to answer any further questions.
>
>
> * [Simulations on real dataset] Many thanks for the questions on simulations.
>
>     - The empirical evaluation for the optimal cycling sampling order, i.e., Fig. 2, was conducted on an artificial dataset for quadratic problems. However, each individual cost function does not share the common minima. In fact, as we showed in Appendix I, there are arbitrarily many instances that do not share the common minima while satisfy the data-heterogeneous condition, i.e.,  $\rho=O(\frac{1}{n})$.
>
>     - The speedup superiority of the optimal cyclic order to the other cyclic orders highly depends on the factor $\rho$ in (21). When $\rho$ is large, the benefits brought by the optimal cyclic sampling becomes less but still exists.
>
>     - We ran the same experiments on real datasets CIFAR-10 and MNIST as that in the left plot in Figure 2, and found the superiority of optimal cyclic sampling still exists, but its speedup effects get less than that presented in the left plot in Figure 2, see more details in the below table (in which cyc. 1-4 are four arbitrary cyclic sampling orders). It is because the heterogeneity indicator $\rho$ for these datasets is no longer on the order of $O(\frac{1}{n})$, but lies somewhere between $O(1/n)$ and $O(1)$. We also fill in the test accuracy for each order and dataset. (Note that we did not tune the hyper-parameter carefully and the test accuracy may not be perfect)
>
>
> CIFAR-10:
>
> | Epoch |0 | 20 | 40 | 60 | 80 | 100 | 120 | test accuracy|
> |:----:|:-----:|:-----:|:-----:|:-----:|:-----:|:-----:|:-----:|:-----:|
> | Cyc. 1 | 0.9443 | 0.3597 | 0.1626 | 0.0986 | 0.0548 | 0.0300 | 0.0091 | 77.46%|
> | Cyc. 2 | 0.9446 | 0.3363 | 0.1583 | 0.0838 | 0.0471 | 0.0274 | 0.0085 |  77.52% |
> | Cyc. 3 | 0.9645 | 0.3400 | 0.1591 | 0.0840| 0.0471 | 0.0274 | 0.0086 | 77.52% |
> | Cyc. 4 | 0.9437 | 0.3488 | 0.1602 | 0.0866 | 0.0486 | 0.0275 | 0.0088 | 77.50% |
> | Opt. Cyc.  | 0.9182 | 0.3048 | 0.1373 | 0.0698 | 0.0376 | 0.0210 | 0.0070 | 77.61%|
>
>
> MNIST:
>
> | Epoch |0 | 5 | 10 | 15 | 20 | 25 | 30 | test accuracy|
> |:----:|:-----:|:-----:|:-----:|:-----:|:-----:|:-----:|:-----:|:-----:|
> | Cyc. 1 | 0.2649 | 0.0009 | 2.2e-5 | 9.6e-7 | 5e-8 | 3.1e-9 | 2.5e-10 | 87.84% |
> | Cyc. 2 | 0.2654 | 0.0010 | 3.2e-5 | 1.4e-6 | 6.9e-8 | 3.6e-9 | 2e-10 | 87.95% |
> | Cyc. 3 | 0.3241 | 0.0008 | 2e-5 | 8.8e-7 | 4.3e-8 | 2.2e-9 | 1.3e-10 | 88.19% |
> | Cyc. 4 | 0.2652 | 0.0011 | 3.5e-5 | 1.5e-6 | 7.5e-8 | 3.9e-9 | 2.2e-10 | 88.10% |
> | Opt. Cyc.  | 0.2097 | 0.0008 | 1.8e-5 | 7.2e-7 | 2.9e-8 | 1.1e-9| 5.2e-11|  88.46%|
>
> **Minor comments:**
>
> * We have put the results in Appendix K to a newly-created Table 1 listed in the rebuttal response Part II to Reviewer y5Mf.
>
> * Sorry we cannot provide plots in this open review system. We have added the final test accuracy in the above two tables.
>
> We will be glad to clarify any further comments. Thanks again!

---

### Decision · Program_Chairs · 2021-09-27

**Decision:**

Accept (Poster)

**Comment:**

This paper is favored by two of four expert reviewers and is acceptable to the other two. Reviewers agree that it is clearly written and organized. There is also general agreement that parts of the analysis are new and insightful, and possibly useful outside of this paper, in particular the "order-specific" norm that is introduced.

Some reviewers make valid points about the limitations of the end result. For instance, reviewer VeN3 points out that the convergence rate is not necessarily encouraging even if it matches plain GD since the paper's method has memory requirements that scale linearly with the dataset size. The relative advantage then happens in an optimistic case where an optimal cyclic ordering is known (in practice, this ordering is replaced by a heuristic) and when the data is sufficiently heterogeneous (a technical condition detailed in the paper).

I will add that the reviews on this paper were quite thorough, and that there was detailed discussion among reviewers and authors. Ultimately, these exchanges addressed many reviewer concerns and left the authors with suggestions for improvement that might only strengthen the writing further. They've even led to some additional empirical measurements and several references to related work.